

# Evaluating the use of smart sensors in ground-based monitoring of landslide movement with laboratory experiments

Alessandro Sgarabotto[1], Irene Manzella[1,2], Kyle Roskilly[3], Miles J. Clark[4], Georgie L. Bennett[5], Chunbo Luo[6], Aldina M. A. Franco[4]

[1]School of Geography, Earth and Environmental Sciences, University of Plymouth, Plymouth, UK
[2]Faculty of Geo-Information Science and Earth Observation, University of Twente, Enschede, NL
[3]Environment and Sustainability Institute, University of Exeter, Penryn, UK
[4]School of Environmental Sciences, University of East Anglia, Norwich, UK
[5]Department of Geography, University of Exeter, Exeter, UK
[6]Department of Computer Science, University of Exeter, Exeter, UK

*Correspondence to*: Alessandro Sgarabotto (alessandro.sgarabotto@plymouth.ac.uk)

**Abstract.** Boulders and cobbles embedded on the body of landslides are carried downstream under the action of gravity, and the study of their transport can give important insight on their dynamics and hence the related hazard. The study examines the reliability of smart sensors to track movements of a cobble and discern between intensity and mode of movement in laboratory experiments. A tag equipped with accelerometer, gyroscope, and magnetometer sensors was installed inside a cobble. The experiments consisted of letting the cobble fall on an inclined plane. By tilting the inclined plane at different angles, different modes of movement such as rolling, bouncing, or sliding were generated. Sliding was generated by embedding the cobble within a thin layer of sand. The position of the cobble travelling down the slope was derived from camera videos. Raw sensor data allowed detection of movement and separation of two modes of movement, namely rolling, and sliding. Additionally, reliable values for the position, velocity, and acceleration were determined by feeding a Kalman filter with smart sensor measurements and camera-based positions. Furthermore, by testing LoRaWAN wireless transmission through sand, the study showed that the signal strength tended to decrease for thicker sand layers. These findings confirm the potential to use these sensors to improve early warning systems and further studies are in progress to assess practicalities of their use in field settings.

**Keywords**: smart sensors, boulders, landslides, physical experiments

## 1 Introduction

The higher frequency and intensity of extreme weather conditions under climate change have increased global landslide hazards (Gariano and Guzzetti, 2016; Sim et al., 2022). At the same time, population growth, land change use and urbanization have increased the vulnerability and the risk related to these events (Pollock and Wartman, 2020). Slow-moving landslides typically occur on high slopes composed of clay-dominated soil with a complex subsurface network sensitive to seasonal rainfall (e.g., De Blasio, 2011; Lacroix et al., 2020). These systems usually move to catastrophic failure by gradually increasing the displacement or alternating seasonal or yearly periods of stability and movement (Saito, 1969; Crosta et al., 2017; Chang and Wang, 2022). The rate of change of displacement can be used to predict the time of failure,





giving reliable results in many field applications (Voight, 1988; Dick et al., 2015; Carla' et al., 2017; Intrieri et al., 2019). To predict the failure of slow-moving landslides, spaceborne and/or ground-based measurements of the activity of the body of the landslide are needed as precursory signals of collapse (e.g., Guzzetti et al., 2020). Ground-based (GB) methods are often used to validate the remote-sensing based approaches and for early warning systems (e.g., Uhlemann et al., 2016). Traditional methods such as the use of extensometer and total station are adopted often in short-duration and low-frequency monitoring campaigns and thus they have limited opportunity of detecting long-term behaviour, understanding the site sensitivity to perturbations, and having spatially well- distributed data (e.g., Dunncliffe, 1988; Crosta et al., 2014). Optical fibers have been recently investigated to measure strain and displacements in the body of the landslide (Schenato et al., 2017). This technology achieves high sensitivity to small strains without being invasive to physical phenomena. However, the field applications require digging trenches in suitable positions on the body of the landslide. Ground-based inSAR (GB-InSAR) is an expensive technology able to achieve high performance in terms of resolution (around 1 millimeter) and temporal coverage (around 10 min) (Pieraccini and Miccinesi, 2019; Kelevitz et al., 2022). Its use becomes more problematic when the vegetation cover is extensive. Moreover, since it is usually installed with a solar panel, the application of GB-InSAR is problematic in regions with limited solar capacity.

Recent advances in Wireless Sensor Network (WSN) and Internet of Things (IoT) technologies, microelectronics and machine learning offer new opportunities to effectively monitor stability of boulders embedded in landslides (e.g., Kumar et al., 2019; Wang C. et al., 2022). Passive Radio Frequency Identification (RFID) tags have been installed in boulders to track their displacements (e.g., Le Breton et al., 2019, 2022). These tags provide good information about the movements with low battery consumption (Le Breton et al., 2019). However, as passive sensors, they require separated antennas to retrieve displacement data from the device. The installation of multiple antennas ensuring all RFID tags have a good connection is problematic due to the site topography and tag distribution. Thus, when the fixed antennas are not in a close range, an operator has to walk on the body of the landslide with a portable antenna to detect the tagged boulders and retrieve the data.

Recently development in the Internet of Things and in wireless communication has allowed the development of compact (few millimeters) and affordable sensors that can measure different environmental features and send the data collected through a wireless network with low power consumption (e.g., Gronz et al., 2016; Dini et al., 2021). The range of environmental features gauged is wide including for example temperature, magnetic field strength, motion, electric currents, pressure, and humidity (Mahmood, 2019). These sensors deployed in geomorphology applications consist usually of Inertial Measurement Units (IMUs) equipped with a transceiver and antenna to send data (e.g., Maniatis et al., 2021). More recently, boulders embedded in the body of a landslide have been used not only in the study of landform shape and evolution, but also in hazard assessment where they can be employed to measure slope-based displacement (Bennett et al., 2016; Shobe et al., 2020; Dini et al., 2021; Shobe et al., 2021; Roskilly et al., 2022, 2023). Sensors embedded in boulders start recording when movement is detected due to rotations or impacts exceeding a custom-defined threshold. Then, the motion data are recorded and transmitted to receivers via LoRaWAN, a wireless communication protocol capable of long distance and low power data



transfer. Finally, the data are transmitted to cloud storage via the internet, usually using cellular networks. Communication is bidirectional, permitting the sensor to both send data and receive messages in return. The versatility and long range of
wireless communication make it easy to get spatially-distributed data from the network of sensors. These are essential features that make the sensors 'smart' (e.g., Le Breton et al., 2022).

Smart sensors were deployed previously in the field to collect data on rock falls (Niklaus et al., 2017; Caviezel et al., 2018, 2019). The sensor was equipped with accelerometers, gyroscopes, and a pressure cell. While the linear and angular motion
were tracked by the accelerometers and the gyroscopes respectively, the pressure sensor better characterised the collisional forces at the point of impact with the ground.  The combination of these three types of sensors allowed measurement of the rock mass falling, rolling, and bouncing, and characterised the rock fall kinematics and the rebound condition on the terrain. This allowed validation of a 3D modelling framework on rock falls (Dorren et al., 2011; Dorren, 2016). Furthermore, smart sensors were used in laboratory experiments to track pebbles embedded in a granular flow (Dost et al., 2020). In this case,
accelerometers, gyroscopes, and magnetometers were installed within five pebbles released in a flume rock avalanche. By detecting the Earth magnetic field, the magnetometers provided information on the direction of the magnetic field and thus contributed to the characterisation of pebble orientation. By combining sensors together and integrating linear accelerations over short time windows, it was possible to derive the trajectories of the pebbles within the granular flow. Although the granular flow was completely contained within the flume, some of the pebble trajectories inferred from sensor data were out
of range as if they fell out of the flume walls.  These small and recent applications highlight that the reliability of these sensors still need to be evaluated for monitoring purposes and tested for the development of early warning systems.

The present study aims to study how different types of sensors (i.e., accelerometer, gyroscope, magnetometer) can be combined, or coupled as described by Dost et al. (2020), to improve the accuracy and reliability of the movement data for
monitoring purposes. Dedicated laboratory experiments were designed to test the potential of the sensor in capturing the magnitude and the mode of movement of a single cobble released down a slope. The sensor was tested over different slope inclinations and modes of movement. The LoRaWAN wireless data transmission was also tested through sand layers of different thickness to study the impact on the signal strength received from the sensor.

This paper is structured as follows.  Section 2 presents the experimental setup and the methodology to retrieve and process
data. Section 3 shows the results from LoRaWAN data transmission tests and the findings on raw and processed data. Section 5 discusses the strengths and the weaknesses of the smart sensor technology in monitoring boulders.  Finally, the paper finishes with some conclusions.



## 2 Material and methods

### 2.1 Experimental setup

The experiments were carried out on a tilt table composed of two panels mounted on a robust metal frame lying on the floor with 6 legs (Figure 1). The first board was 150 cm wide and 150 cm long, the second board was 150 cm wide and 200 cm long. The square board was hinged to the rectangular one along the shorter side. At the opposite side of the square board, an electric rope hoist allowed for lift and tilt of the square board at different angles. The cobble travelling down the slope was released from an acrylic box (30 cm x 43.5 cm x 32.5 cm) installed at the upper end of the table which is closed with a sluice

gate (Figure 1). To control the spreading of the run out when the cobble is embedded in a sand layer, two acrylic side walls were mounted on the table constraining the maximum spread to 30 cm. The experimental setup was as in Manzella et al. (2016) and Makris et al. (2023). Experiments were recorded by a GoPro HERO8 framing the tilting table frontally (Figure 1). The GoPro camera was paired and controlled remotely via Wi-Fi to ensure synchronised recordings. To track moving objects, the GoPro camera was set at 4K resolution with linear distortion and a frame rate of 30 fps. The cobble used in the

experiments has an approximately rounded shape. On the vertical plane the cobble had a diameter of 10 cm, and the cobble thickness was around 8 cm on the transversal plane (Figure 2a and b). The cobble had a cylindrical borehole with a diameter of 4 cm and a depth of 7.5 cm (Figure 2a and b). The hollowed cobble was made of concrete, cast using a mould created from a real cobble. The preparation procedure is described in detail in Clark (2023). A smart sensor was placed within the borehole to detect the movements of the cobble (Figure 2c, d, and e). Specifically, the smart sensor used was the mini-GPS

Tracker (Miromico manual, 2020a, b) which is a versatile and compact device equipped with a 9-axis sensor comprising accelerometers, gyroscopes, and magnetometers (ST LSM9DS1). The device has an additional low-power accelerometer sensor (ST LIS2DH) that monitors motion continuously. This wakes the 9-axis sensor and begins recording when the movement exceeds the user-defined threshold (configured to 390 mg in this study). The device is powered by a 3.6 V lithium thionyl chloride battery. It is also equipped with LoRaWAN wireless communication and a GPS receiver (this was

deactivated for indoor experiments). A microcontroller regulates the data acquisition, processing, and transmission. The sensors in the LSM9DS1 record with their maximum range, i.e. the accelerometers at $\pm16$g, the gyroscopes at $\pm2000$ °/s, and the magnetometer at $\pm16$ Gauss. The sensor tag and the battery were placed within a sensor enclosure stuffed with cotton-pads to damp and prevent damage due to violent impacts (Figure 2c and d). The sensor enclosure was then inserted into the cobble so that the sensor tag sat at the bottom of the borehole. The upper part of the cavity was then sealed with blue tack

(Figure 2e). Once sealed, the tagged cobble weighed 0.7 kg.

The data recorded can be either downloaded manually via USB cable or sent wirelessly to a cloud server for online retrieval via a LoRaWAN gateway. The gateway used was a Dragino LPS8 Gateway placed at a distance of 2-3 m from the experimental table. The gateway runs open-source software based on OpenWRT, with minor customisations of the stock firmware for additional features such as remote access and cellular modem support. The network comprises the LoRaWAN

gateway, and the sensor as an end-node device. The data received by the gateway are relayed to a LORIOT network server



and onwards to the SENSUM cloud server (Roskilly et al., 2022, 2023). In addition to the transmitted data, received signal quality metrics are made available by LORIOT for each transmission, specifically the Received Signal Strength Indicator (RSSI) and the Signal-to-Noise Ratio (SNR). RSSI is a measure of the power of the signal received from the end-node by the gateway and ranges usually between -120 dBm, a weak reception, and -30 dBm, a strong reception. SNR is the ratio of the

power of the received signal to the power of the background signal noise and values are usually between -20 dB and 10 dB, with higher values representing less corrupted signal.

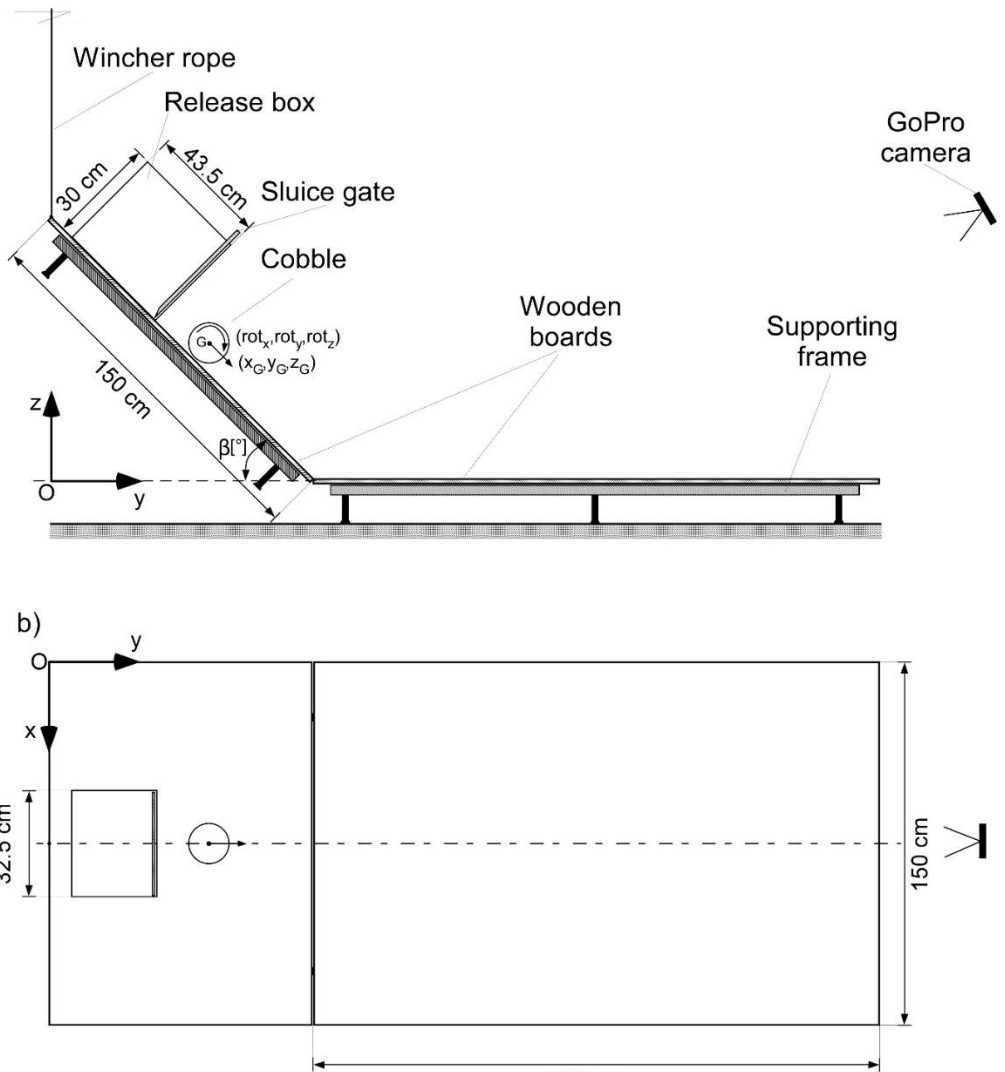

**Figure 1. Experimental setup as in Manzella et al. (2016) and Makris et al. (2023). (a) Lateral view. (b) Top view.**




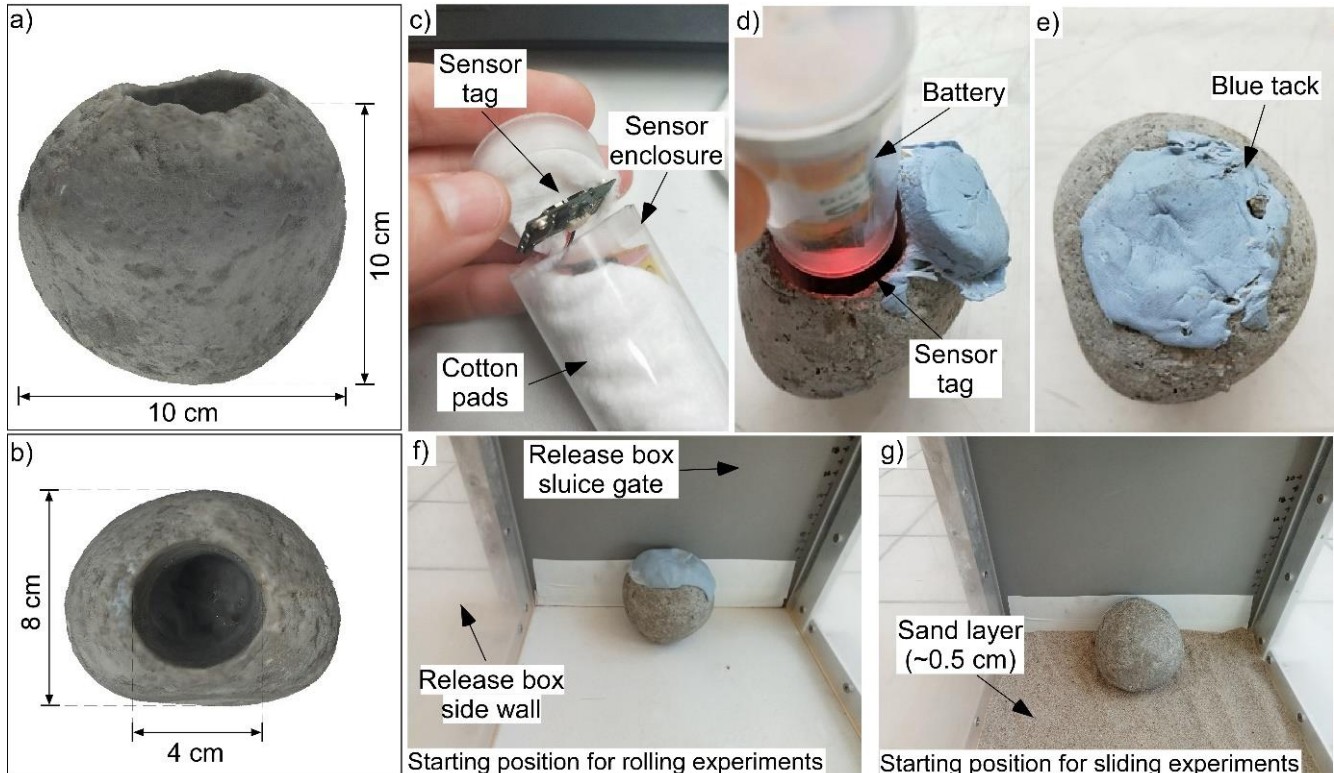

**Figure 2. Laboratory experiments pictures. (a, b) Cobble rendering. (c, d, and e) Sensor installation in the cobble. The starting position of the cobble within the release box for (f) rolling experiments and (g) sliding experiments while embedded in a thin layer of sand.**

## 2.2 Camera tracking


Different techniques have been developed and deployed to detect and track objects over time (e.g., Soleimanitaleb et al., 2019). Classical approaches detect objects by subtracting them from the background when the minimum number of pixels for an object is exceeded (Soeleman et al., 2012; Stolle et al., 2016). In the last 10 years advanced techniques have been
developed to train models to detect objects with specific shapes (Liu et al., 2020). These methods based on deep learning are usually not affected by the light condition and scale of the object to track. Object detection can be improved when the objects to track are known, avoiding background subtraction algorithm. This improvement can be achieved using YOLOv5, a deep-learning algorithm (Redmon et al., 2016).

First, different training images of the object were collected. The images were the ground truths to train the model. In each training image, the object was framed by drawing a bounding box and labelling it with the class name. Third, the set of images was split: 70% for training, 20% for validating and 10% for testing. Cross-validation protects against overfitting in a predictive model, particularly in a case where the amount of data may be limited. Finally, the YOLOv5 Detector was trained over a sample of the data collected. The camera calibration was performed in two steps starting from the original frame



(Figure 3). Second, the camera distortion was reduced following the checkerboard method (Zhang, 2000) using the corresponding built-in functions in the OpenCV computer vision library (Bradsky, 2000). The perspective distortion was then reduced using a suitable built-in function also from OpenCV. Given four corners of the region of interest, the perspective transformation re-projected them in the undistorted reference system where the frame width and frame height were specified to cover the region of interest (namely, the titling table boards). The aspect ratio of the frame (expressed in

pixels) was specified to match the aspect ratio of the board on which cobble motion occurred This procedure allowed the position of the centre of the bounding box to be computed and used as a proxy of the position of the centre of mass for the cobble. The vertical elevation was inferred from the coordinates of the centre of mass of the cobble assuming that it always kept a point of contact on the tilting table.

**2.3 Data-fusion approach**

Micro-electromechanical systems (MEMS) usually suffer from stochastic and deterministic errors, including bias, scale factor error, misalignment, noise, latency, and temperature dependence (e.g., Clemente et al., 2021). These factors hamper tracking based on Inertia Measurement Unit (IMU) and magnetometer (i.e., inertial navigation or dead reckoning), which is

detrimental to its use in position/velocity monitoring. To avoid these issues and harness the best information about object motion, it is necessary to combine sensor data together and use the camera tracking positions as movement constraints (e.g., Dewhirst et al. 2016). This data fusion approach can be divided into five steps as illustrated in the flowchart in Figure 3.

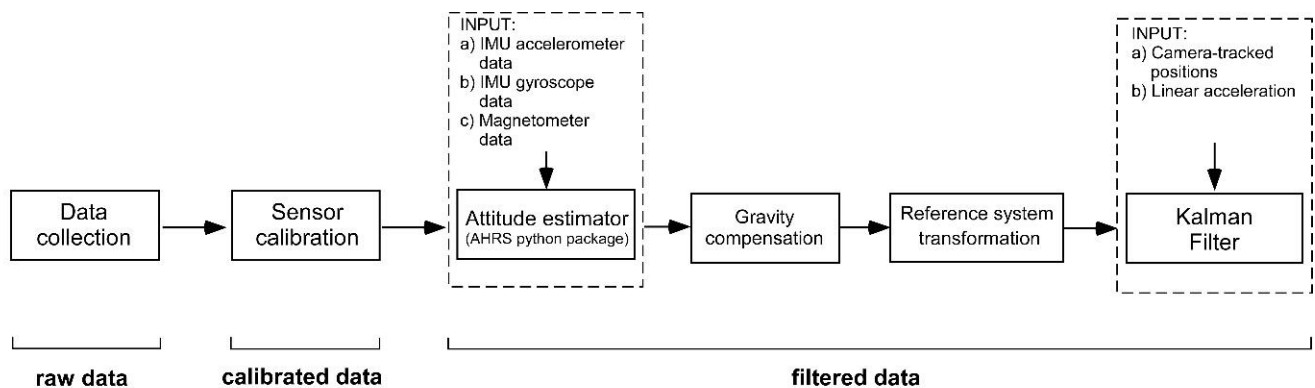

**Figure 3. Pipeline for IMU and camera tracking data fusion. In short, the raw data retrieved from the sensors are calibrated following the procedure related to each sensor and then filtered to better describe cobble kinematics.**

In the first step, the raw recordings are retrieved from the sensor at the end of each run. The second step consists of deriving the calibrated data from the sensor raw recordings. Magnetometers and each sensor in the Inertial Measurement Unit is calibrated. Specifically, the accelerometers are calibrated according to Frosio et al. (2009), the gyroscopes according to

Glueck et al. (2013) and the magnetometers according to Dewhirst et al. (2016). Details on the calibration framework used are reported in the Supporting Information (SI). Third, the orientation is computed combining the readings of the



accelerometers, gyroscopes, and magnetometers (e.g., Madgwick et al., 2011; Mahony et al., 2012; van Zundert et al., 2013).
Here, the orientation is derived following the approach proposed in Mahony et al. (2012) coded in the AHRS (Attitude and
Heading Reference System) python library (AHRS library, 2019). The fourth step is gravity compensation which consists of
adding gravity to obtain linear accelerations from the accelerometers. The fusion of these data sources eliminates the low
frequency drift caused by integration of gyroscope errors, while giving better high frequency accuracy than accelerometer
and magnetometer measurements alone. Given the sensor orientation (attitude) with respect to the local Earth reference
frame, the calibrated accelerometer measurements are rotated from the local body reference frame to the local Earth
reference frame using simple transformations. The last step in the pipeline for IMU and camera tracking data fusion is a
linear Kalman filter fed by the sensor-derived (linear) accelerations and camera-based positions (Dewhirst et al., 2016; Kim
and Bang, 2019). The sensors installed within the cobble provide internally-measured acceleration data, that can be
integrated for velocity and position change but the derived data are subject to drift with time. On the other hand, the camera
provides position data measured externally. It is subject to noise when differentiated for velocity and acceleration but can
constrain the drift from sensor data alone. By combining the measurement data, a two-step recursive algorithm computes the
state of the system defined as the position, velocity, and linear acceleration of the cobble. The uncertainty in the state
estimate is assessed by the relative weight given to the measurements and current state estimate. Further details on the
Kalman filter implementation are present in the Supporting Information.

## 2.4 Design of the experiments

Smart sensors allow recording of motion data and transmission of data through a wireless network. Thus, two experimental
campaigns were carried out to test the performance of smart sensors in monitoring the movements of a cobble. The first set
of experiments was dedicated to studying the LoRaWAN wireless data transmission between a gateway and a sensor used as
an end-node device. The experiments were carried out indoors with the gateway mains powered and the sensor battery
powered. Specifically, the study consisted of measuring the signal strength of data packets transmitted via LoRaWAN when
the device was covered by sand layers of increasing thickness, namely 0 cm, 5 cm, 8 cm, and 10 cm. For these experiments,
the acquisition frequency for the accelerometers and gyroscopes was set at 14.9 Hz, the magnetometers recorded at 5 Hz.
The second experimental campaign was designed to investigate the modes of movement recorded by the sensors installed in
a cobble travelling down different inclines, namely 18°, 25°, 30°, 35°, 40°, 45°, 50° and 55°. The cobble was placed within
the release box with the sealed borehole facing upwards and a side leaning on the sluice gate (Figure 2f). The starting
position allows the cobble to roll down the table. For slopes of 25° to 40° the same experiments were also carried out
embedding the cobble in a thin sand layer (~0.5 cm) to reduce the friction on the table. The cobble was placed within the
release box with the sealed borehole leaning on the sluice gate (Figure 2g). The starting position of the cobble and the sand
layer prevents full rotations and allows the cobble to slide down the table. Each run was repeated three times. For these



movement experiments the acquisition frequency for the accelerometers and gyroscopes was set at 59.5 Hz and the magnetometers recorded at 5 Hz. In this set of experiments, the data were downloaded from the sensor via a USB cable.

## 3 Results

### 3.1 Received signal transmitted by LoRaWAN through a sand medium

The sensor communication system was tested with different thicknesses of sand coverage to reproduce the attenuation that the terrain may exert on the signals when a smart boulder is installed on the body of a landslide. A plastic box containing the sensor tag and the battery was placed in a bucket. The plastic box was covered by sand so that a minimum sand layer was achieved in all directions. Then, by moving the bucket, the sensor recorded movement and sent data to the gateway, which was placed at a distance of 2-3 m. The signal strength was tested for different sand layer thicknesses of 0 cm, 5 cm, 8 cm, and 10 cm. The power of the received sensor data transmissions was evaluated with two metrics, the Received Signal Strength Indicator (RSSI) and the Signal-to-Noise Ratio (SNR) (Figure 4). For the accelerometer and gyroscope data packets, the median value of RSSI is -35 dBm when no sand covers the sensors (Figure 4a). The median value decreases to -45 dBm when sensors are submerged in a 10-cm sand layer. Moreover, the cumulative probability increases its slope suggesting data closely distributed around its median value. Similarly, the signal strength indicator of the magnetometer data packets is sensitive to sand layer depths (Figure 4b). Indeed, the median value RSSI decreases from -35 dBm to -55 dBm. In contrast with the behaviour of accelerometer and gyroscope packets, the cumulative probability becomes less tilted since the sample is more dispersed around the median. Overall, the RSSI is more sensitive to sand submergence with the magnetometer data than with the gyroscope and accelerometer data. However, this sensitivity behaves differently. For the accelerometer and gyroscope packets, the RSSI becomes concentrated around the median value. On the other hand, for the magnetometer packets, RSSI is lower but more dispersed when the sensor is submerged in a sand layer probably due to a different acquisition frequency respect to the gyroscope and accelerometer. Moreover, the SNR is not as sensitive as RSSI to sand layer submergence. SNR is positive ranging between 5.4 dB and 13.5 dB for the accelerometer and the gyroscope data packets (Figure 4c) and between 5.2 dB and 13.0 dB for the magnetometer data packets (Figure 4d). Regardless of the sand thickness tested, the median value for SNR stays approximately constant for both packet types. Since SNR drops are usually related to changes in environmental factors such as humidity and temperature (e.g., Jeftenić et al., 2020), this finding seems to support that the laboratory conditions remained the same during the experiments







**Figure 4. Cumulative probability distribution of (a and b) RSSI and (c, d) SNR when the signal is transmitted through different sand layer depths, namely 0 cm, 5 cm, 8 cm, and 10 cm. (a, c) Accelerometer and gyroscope packets. (b, d) Magnetometer packets.**

### 3.2 Raw data from accelerometers, gyroscopes, and magnetometers


Raw data recorded on a 30˚ incline are shown as an example of characteristic signals of the sensors while the cobble travels down the slope (Figure 5). Similar observations can be derived from the raw data collected on other slope angles. Sensor outputs show different ranges when the cobble freely travels down the tilt table (Figure 5a, b, c) compared to when it travels while embedded in a sand layer (Figure 5c, d, e). Following its release, the cobble was seen simply sliding down the incline

when embedded in the sand, but without the sand it rolled down the table instead. These different modes of movement are evident in the gyroscope recordings. When the cobble is not embedded in a sand layer, the angular velocities increase progressively reaching maximum magnitude at the junction between the sloping and the horizontal board and then gradually decrease (Figure 5a). The angular velocities along all axes range between -2000 ˚/s and 1200 ˚/s, confirming the rolling





movements seen in the experiments. Conversely, when the cobble is embedded in a sand layer, rotations are limited and thus
the angular velocities show less variation, between -200 °/s and 400 °/s (Figure 5d). The smaller range for the angular velocities denotes a different behaviour, confirmed by the sliding observed during the experiment. The accelerometer detects the impacts that occur during the motion. When the cobble rolls down the slope, the acceleration signal shows spikes across all axes that represent the contact impacts between the cobble and the experimental table during the rolling motion (Figure 5b). The highest magnitude peaks occur when the cobble touches the horizontal board at the end of the sloping board and
reaches the value of -14.5 g in the vertical direction. Conversely, when the cobble is embedded in a sand layer, the accelerometer signals are smoother ranging between –0.5 g and 2.0 g (Figure 5e). Accelerations in all directions show smaller changes with a small spike occurring when the slope become horizontal along the experimental table. Some differences also emerge from the magnetometer recordings. In the sliding experiment, the magnetic field signal is flat in all directions as the cobble travels down the slope and it changes slightly as the cobble slows down on the horizontal board
(Figure 5f). Similarly, in the rolling experiments, while the cobble is on the slope the magnetic field data are approximately constant with time and are then subject to larger variations when the cobble decelerates (Figure 5c). Between modes some differences are seen in the magnetic field values, ranging between -0.5 and 0.5 Gauss for the rolling experiments and -0.2 and 0.6 Gauss for the sliding ones.





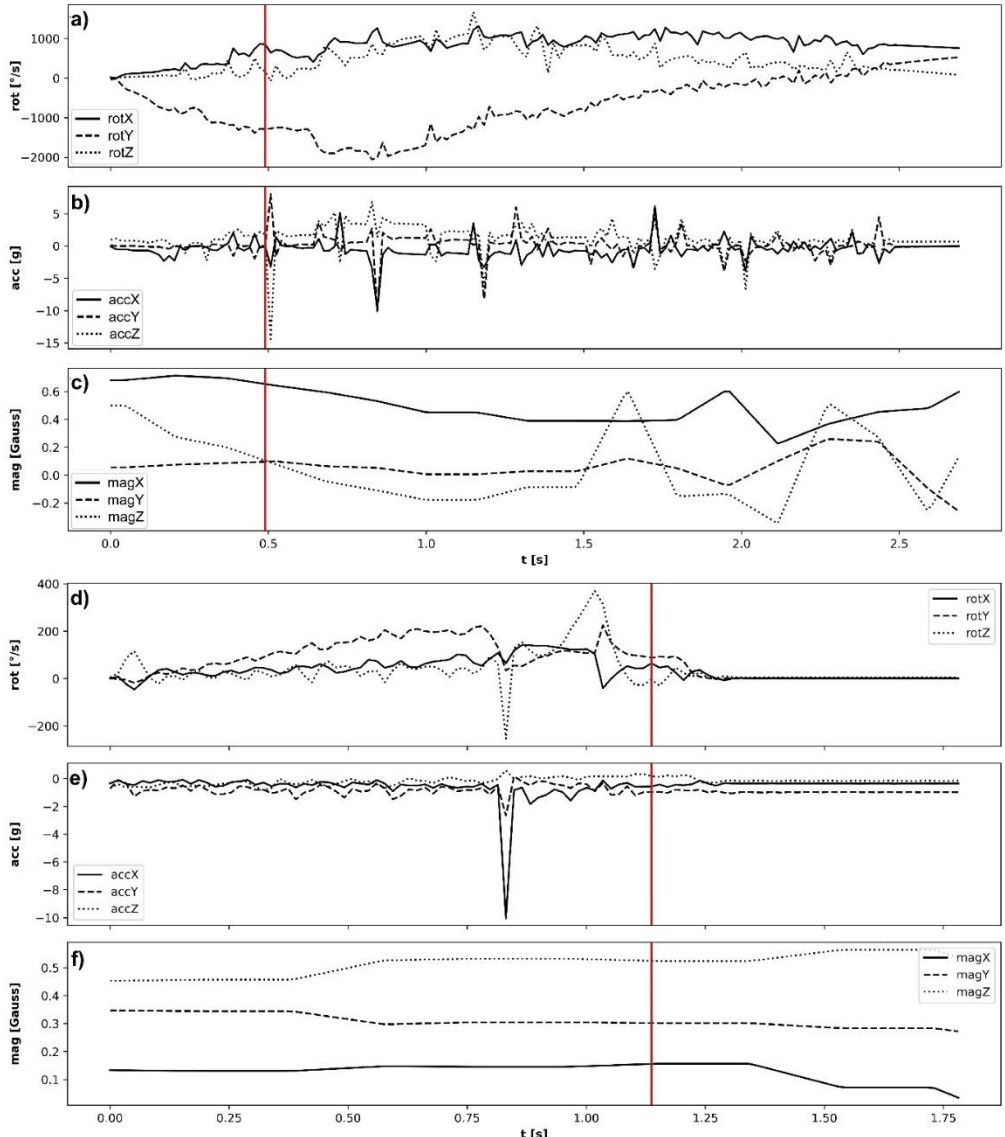

**Figure 5. Raw recordings of the three sensor types on a 30° incline for (a, b and c) a rolling experiment and (d, e and f) a sliding experiment. (a, d) gyroscopes data, (b and e) accelerometers data. (c, f) Magnetometers data after upsampling. The solid line refers to the x axis, the dashed line to the y axis and the dotted line to the z axis. The solid red line shows the time when the cobble passes over the slope break.**

### 3.3 Camera-based positions

The cobble paths tracked by the camera on the 30° incline are shown to highlight the differences between rolling trajectories (Figure 6a) and sliding trajectories (Figure 6b). When the cobble is rolling, the trajectory keeps approximately straight and then, in the second half of the horizontal board, tends to drift to the left-hand side (test 3) or to the right-hand side (tests 1 and 2). These paths can be explained based on the momentum and the irregular shape of the cobble. After the release, the



cobble accelerates rapidly increasing its momentum. Thus, along the slope, when the momentum is high, the cobble hardly drifts from its initial direction. At the junction between the slope and the horizontal board, the cobble keeps on rolling in the same direction without losing contact with the surface. In the second half of the horizontal board, when the momentum has reduced, the irregular shape of the cobble makes its trajectory drift. Conversely, when the cobble is embedded in sand, it keeps approximately the same side in contact with the experimental table and slides down the slope. Since rotations are

limited, the irregularities on the cobble surface only slightly affect the trajectory so that it is approximately straight. On the horizontal board, the cobble motion is quickly stopped by the sand layer. Thus, the resulting trajectories are shorter than in the rolling experiments.

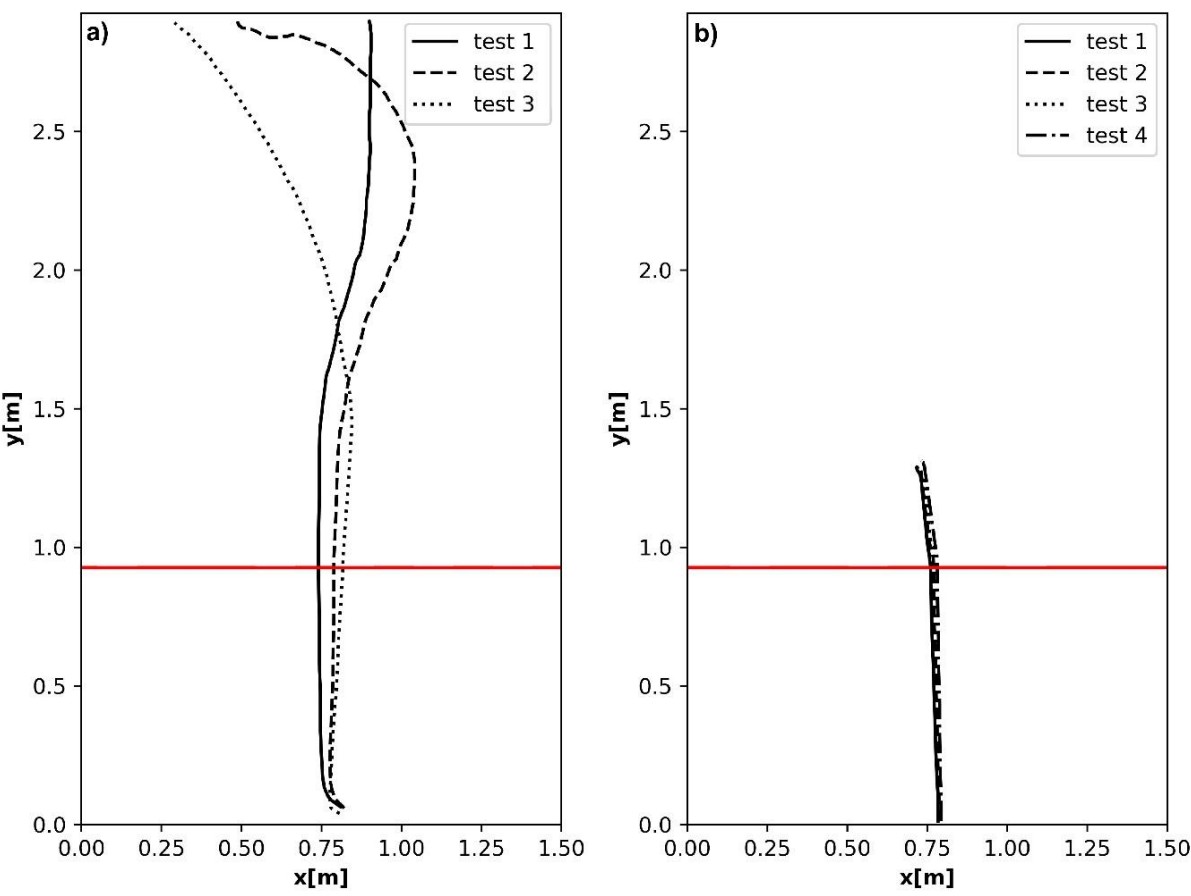

**Figure 6. Trajectories extracted from camera videos for (a) rolling tests and (b) sliding tests carried out on a 30° incline. The rolling tests were repeated three times, the sliding tests were repeated four times. Each run is denoted by the different line styles. The solid red line defines the position of the slope break.**

**3.4 High-level representation of raw data**




The raw data collected describe the cobble movements in space and time. To characterise the mode of movement for each inclination, a typical behaviour was inferred from the data following the procedure described. First, the vector magnitudes of the acceleration, angular velocity and the magnetic field data was computed. Second, each vector norm was averaged over the run duration. Then, boxplots were computed from these averages. The cobble activity was grouped into boxplots
representing the spatiotemporal averages over all runs for each slope and for rolling and sliding experiments (Figure 7). In Figure 7, boxplots on the left-hand side refer to motion on the inclined plane (Figure 7a, c and e), and boxplots on the right-hand side correspond to the motion on the horizontal plane (Figure 7b, d and f).

The boxplots referring to the gyroscopes show a clear separation between rolling and sliding experiments from both magnitude and distribution of angular velocity (Figure 7a, b). Specifically, sliding experiments exhibit similar values for
angular velocity on the inclined plane (Figure 7a) and on the horizontal plane (Figure 7a) regardless of the slope angle (rot≈0°/s). Conversely, rolling experiments show different behaviour depending on inclination. On the inclined plane, the angular velocity becomes larger as the tilt table increases to 50° and then drops at a slope of 55° (Figure 7a). A similar trend is shown on the horizontal plane where the cobble released from a more inclined plane has larger values of angular velocity (Figure 7b). On the horizontal plane, at around 50°-55 °, the increasing trend of angular velocity also shows a sudden drop on
the horizontal plane. We speculate that for higher slopes, the cobble starts losing contact with the experimental table and thus the angular velocity is slower. On both planes, the angular velocity for rolling experiments range between 1000 °/s and 2500 °/s.

Raw acceleration data shows a clear separation between rolling and sliding experiments (Figure 7b, c). This separation
suggests a remarkable difference in the acceleration depending on the mode of movement, as the acceleration in sliding experiments (1-1.5 g) were smaller than in rolling experiments (namely 1.5-4.0 g). Since the sensor enclosure was not perfectly in a fixed position within the borehole, the vibrations were seemingly higher than in the rolling experiment rather than in the sliding experiments resulting in higher values of total acceleration.

On the inclined plane, the acceleration does not increase with the slope inclination and shows an irregular trend. Specifically,
the acceleration increases up to a 35° incline, then it decreases at 40°, it goes up again at 50° before dropping at 55°. On the horizontal plane, the acceleration in rolling experiments is more regular as the cobbles released from higher slopes show larger acceleration (Figure 7d). However, between 50° and 55°, the acceleration drops stopping the increasing trend. Although the raw acceleration magnitude and distribution separates the modes of movement the irregular behaviour, especially for high slopes, is not completely clear from a physical point of view. Raw acceleration data need further
investigation to explain the cobble dynamics on the slope.

Similarly, the raw magnetometer data show irregular behaviour. Raw magnetometer data have a similar range in sliding and rolling experiments (Figure 7e, f). However, in the rolling experiments, the magnetic field magnitude decreases nonmonotonically by increasing the inclination up to 35° on the inclined and the horizontal plane. For a slope of 40°, the


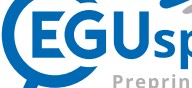

magnetic field drops abruptly and then shows similar values for higher slopes on both the inclined and the horizontal plane. Conversely, in the sliding experiments, raw data increases with incline up to 35° and then it slightly decreases at 40°. Overall, the magnetic field magnitude changes nonmonotonically for rolling and sliding experiments. However, in the rolling experiments, the magnetic field tends to decrease as the slope angle increases, whereas, in the sliding experiments, it has the opposite trend. The same behaviour is observed on both the horizontal and the inclined plane. Although different trend

responses are seemingly detected for a slope increase, the magnetometer does not allow characterisation of the mode of movement in its own right.



**Figure 7. High-level representation of cobble motion for rolling and sliding experiments using raw data. (a, c, and e) Boxplots for motion on the inclined plane. (b, d, and f) Boxplots for the motion on the horizontal plane. The mean magnitude of (a and b) angular velocity, (b and c) acceleration, and (e and f) magnetic field. White boxplots refer to experiments with a sliding cobble, whereas grey boxplots are tests with a rolling cobble.**

## 4 Discussion

### 4.1 Kinetic energy as representation of motion

The total kinetic energy of an object is defined as the sum of the translational and rotational energy (e.g., Díaz, 2019):

$$E_{TOT} = K + R = \frac{1}{2}mv_G^2 + \frac{1}{2}\omega^T I \omega \qquad (1)$$

where $m$ is the mass of the cobble, $v_G$ is the velocity magnitude of the centre of mass, $\omega$ is the angular velocity, while $I$ is the moment of inertia of the cobble. By approximating the cobble as a sphere of radius R, the moment of inertia is $\frac{2}{5}mR^2$. The first term on the right-hand side is the translational energy, whereas the second term is the rotational energy. Given the mass and the average diameter of the cobble, it is possible to compute the variation of the average total kinetic energy with respect to the inclination of the slope using the magnitude of linear and angular velocity (Figure 8).

For the rolling experiments, on the inclined plane, the total kinetic energy increases with incline until it reaches 45° (Figure 8e). The energy slightly decreases at 50°and 55°, where the average value stays between 4 J and 6 J. Similarly on the horizontal plane (Figure 8f), the total kinetic energy increases with the slope angle but suddenly decreases when the slope angle reaches 50°- 55°. For sliding experiments, on the inclined plane, the total kinetic energy increases until the slope reaches 35° and then it reduces (Figure 8e). On the horizontal plane, the drop in the total kinetic energy occurs at a slope angle of 35 ° - 40° (Figure 8f). A better understanding of the total kinetic energy is provided by the translational and rotational energy. In rolling experiments, as the slope inclination increases, the rotational energy increases and suddenly drops at 55° both on the inclined and horizontal plane (Figure 8c, d). The translational energy increases monotonically on the horizontal plane (Figure 8b). Conversely, on the inclined plane, the increasing trend of the translational energy is interrupted with sudden drops at 40° and at 50° -55° (Figure 8a). The cobble makes small bounces when it hits the horizontal board. Consequently, the cobble does not keep a point of contact with the boards while it transits from the tilted board to the horizontal board. The impact dissipates energy leading to its decrease at 50° and 55°. In sliding experiments, the rotational energy is very small so that there is a clear separation between modes of movements on the inclined and horizontal planes (grey and white boxplots in Figure 8c, d). The translational kinetic energy on both planes is lower than in the rolling experiments but has the same trend for slope changes between 30° and 40° (Figure 8a, b). Comparing rolling and sliding tests on the same slope, the average value of the total kinetic energy is slightly larger in the rolling experiments since the rotational energy is higher than in the sliding experiments. Indeed, in sliding experiments, the rotational kinetic energy is





approximately constant and smaller than in rolling experiments on the inclined and horizontal planes. The translational
kinetic energy in sliding experiments has a trend similar to the rolling experiments but with slightly smaller values. Despite
the IMU and camera recordings fusion, the variation of the kinetic energy for change in slope is not fully explained by the
cobble dynamics probably because the sensor tag within the cobble is not completely secured.



**Figure 8. Boxplots of (a, b) the mean translational kinetic energy K, (c, d) rotational kinetic energy R and (e, f) total kinetic energy**
**$E_{tot}$ in rolling and sliding experiments. (a, c, and e) Motion on the inclined plane. (b, d, and f) Motion on the horizontal plane.**
**White boxplots refer to experiments with a sliding cobble, whereas grey boxplots describe tests with a rolling cobble (m =0.7 kg; R**
**= 0.05 m; I = 1.68 kg·m$^2$).**

## 4.2 Validation






To assess the robustness, sensors data have been compared to conceptual models that describe the process according to physical laws (Figure 9). Despite the motion conceptualisation, the analytical models describe the essential motion features expected to be captured by the sensor. Given the simplification introduced, the comparison between experimental results and model predictions considers only some of the variables retrieved from the sensor data specified in the following subsections.


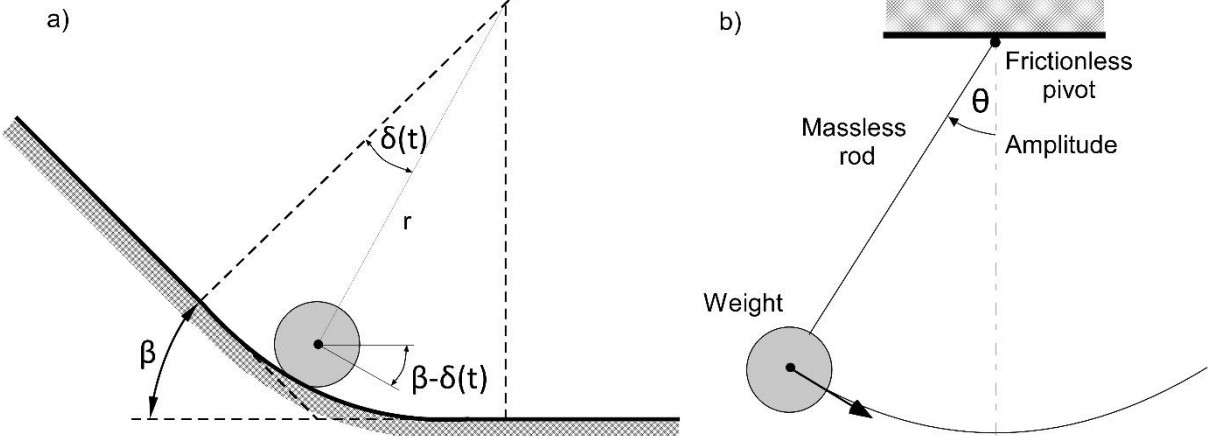

**Figure 9. Simple conceptual frameworks to assess the robustness of the sensors data. (a) Rigid body motion over a tilt plane. (b) Pendulum oscillations around a frictionless pivot.**

### 4.2.1 Simple rigid body motion over a tilt plane


A rigid body travelling down a tilt plane is subjected to different forces, namely gravity, friction, and centrifugal force in a high-curvature path. Assuming the rigid body as a point-like mass, these forces induce an acceleration that can be defined as (Manzella and Labiouse, 2013)

$$a_{t+1} = g \sin(\beta - \delta_t) - \tan \phi \cos(\beta - \delta_t) - \frac{v_t^2}{rg} \tag{2}$$

where g is the acceleration due to gravity, $\beta$ is the tilt plane slope, $\Phi$ the dynamic friction angle, and $r$ is the radius of curvature. These parameters characterise the tilt plane and its surface. When the path slowly increases its curvature (i.e., $\frac{1}{r} \neq 0$ ), the direction of friction and weight forces change with angle $\delta_t$ (Figure 9a). By zero-order hold integration with respect to time, the velocity $v$ and position $s$ are

$$v_{t+1} = v_t + a_{t+1} \Delta t \tag{3}$$

$$s_{t+1} = s_t + v_t \Delta t + \frac{1}{2} a_t \Delta t^2 \tag{4}$$

The framework refers to the kinematics of the centre of mass in the direction of the movement. Thus, the theoretical model is
used to validate the position, velocity, and acceleration along y and z directions. The modelling parameters are adjusted to





improve the comparison between the modelling and experimental results. The typical experimental behaviour was inferred from repeats by computing the average and the standard deviation of position, velocity, and acceleration at each time step along y and z directions. Thus, in the following plots, the black solid line represents the average behaviour, whereas the grey area shows the standard deviation of the variable. For sake of simplicity, the comparison between experiments and modelling

results is shown for the experiments carried out at a slope of 30° (Figures 10 and 11). Further comparisons between the modelling framework and the data collected at different slopes are available in the Supporting Information.

The trajectories of cobbles travelling down the slope are represented in Figure 10. For the rolling experiments, the cobble average trajectory is smooth showing a larger variability on the slope (Figure 10a). Then, when the cobble is on the

horizontal board, the trajectory has a low variation from the average showing some wiggles due to the uneven cobble surface that makes the rolling motion irregular (Figure 10a). Conversely, in sliding experiments, the cobble trajectory is smoother along the whole path (Figure 10b). Similarly to the rolling case, the variability of the cobble path is smaller on the horizontal board and is slightly larger on the slope. In both examples, the modelling framework describes a path that matches approximately the experimental trajectories.


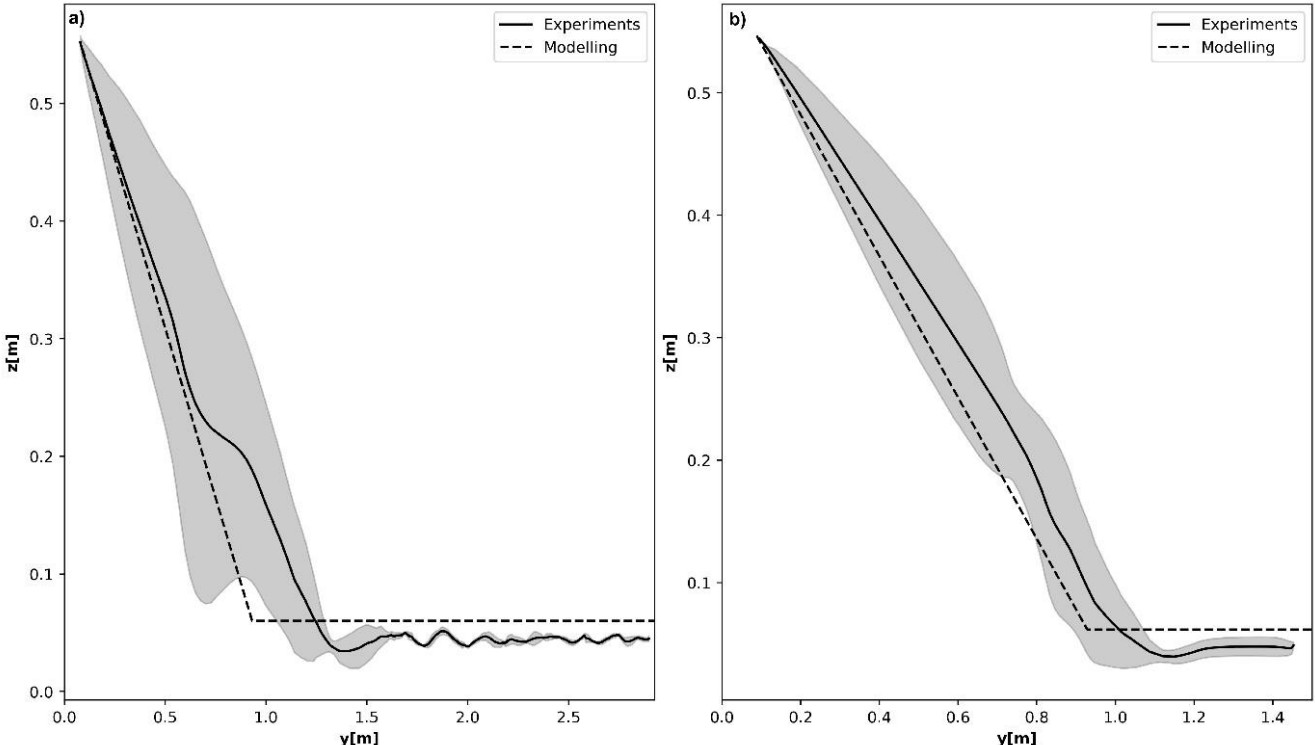

**Figure 10. Comparison between the experiments and the analytical model describing the trajectories of the cobble for a slope at 30° in the vertical plane z-y. (a) Rolling experiments where the response of the conceptual model is obtained for $r$ =0 m and $\Phi$ = 7.5 °. (b) Sliding experiments where the response of the conceptual model is obtained for $r$ =0 m and $\Phi$ = 2.5 °. Solid lines represent the**



**435** **average behaviour during the experiments at the same slope, whereas the grey area show the standard deviation of the variable considered.**

In sliding experiments, as the cobble travels down the slope under gravity, there is an acceleration phase where the horizontal and vertical velocity increase their magnitude reaching a peak at the end of the slope. Then, on the horizontal board, there is a deceleration phase where the friction prevails decreasing the cobble velocity. The modelling predictions are

**440** computed assuming a friction angle of 7.5˚ and assuming a curvature $\frac{1}{r}$ equal to zero since the flat boards of the experimental table have infinite radius. No curvature is assumed also in the joint between the two flat boards. Compared to experimental results, the theoretical velocity changes more slowly so that the velocity peaks are delayed by approximately 0.5 s in the vertical and in the horizontal direction (Figure 11a and c). The averaged experimental velocity in the vertical and horizontal direction shows wiggles due to the uneven surface of the cobble. The two acceleration phases can be made out from the

**445** averaged acceleration (Figure 11e and g). The horizontal acceleration quickly decreases oscillating around zero. The vertical acceleration increases and oscillates around zero. The acceleration along the vertical and the horizontal directions gets to zero with a delay as pointed out for the velocity. Conversely in sliding experiments, when the cobble is embedded in a sand layer, the predictions of the conceptual model are poorer (Figure 11b, d, f, and h). The modelling predictions are computed assuming a friction angle of 2.5˚ and neglecting the effects of the curvature of the experimental table. When the cobble is on

**450** the slope, the experimental values for the velocity and acceleration are different from modelling predictions. On the horizontal board, these differences become smaller. In this case, when the cobble is embedded in sand, the interaction forces between the sand layer and the cobble are neglected in the modelling framework and thus its prediction becomes poorer. The experimental results show some differences that can be related to the highly simplified conceptual model and the limitations of the experimental setup, such as the non-perfectly fixed position of the sensor tag and the irregular shape of the cobble.

**455** Despite these differences, the range of accelerations and velocity computed by coupling the sensors and the camera is well-approximated by the conceptual model.



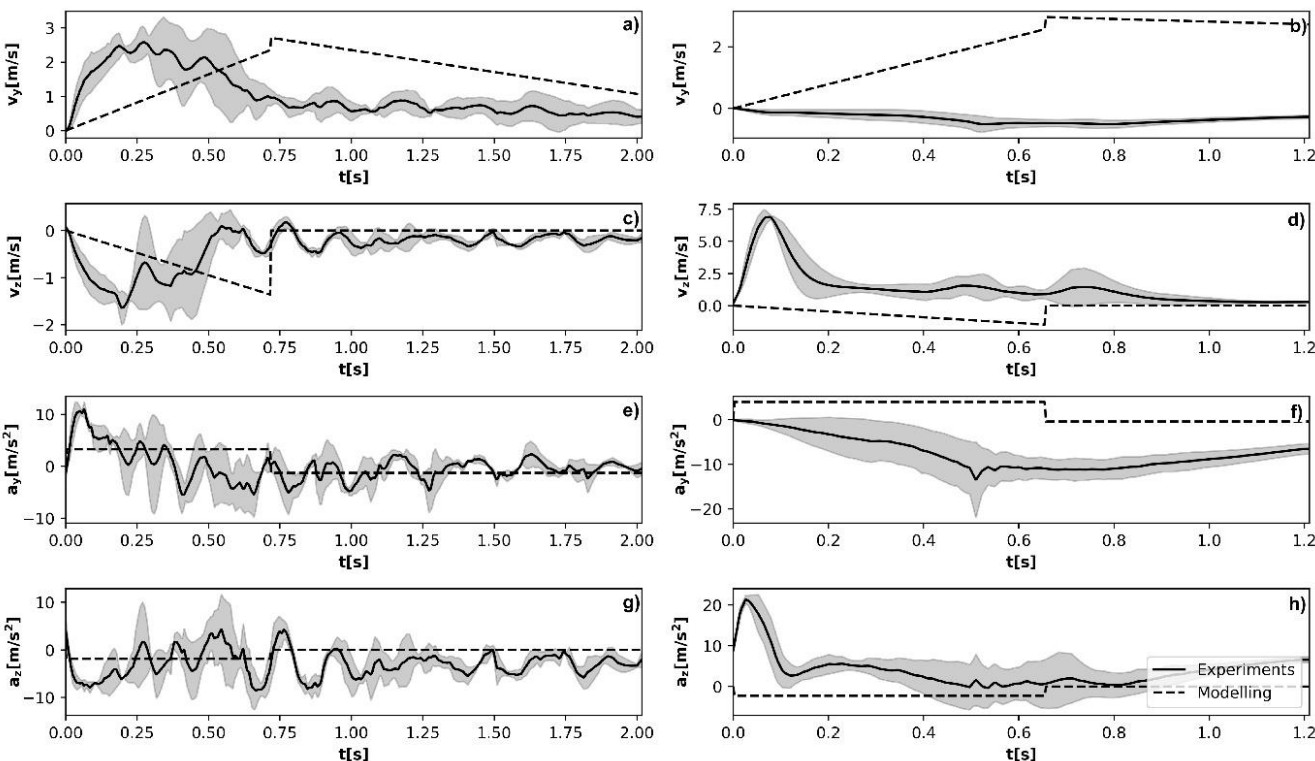

**Figure 11. Comparison between the experimental results and the analytical model for (a, c, e and g) rolling and (b, d, f, and h)**
**sliding on a 30˚ incline. (a, b) Horizontal velocity $v_y$. (c, d) Vertical velocity $v_z$. (e, f) Horizontal acceleration $a_y$. (g, h) Vertical**
**acceleration $a_z$. Dashed lines show the response of the conceptual model obtained with $r$ =0 m and $\Phi$ = 7.44˚ for the rolling**
**experiments and $r$ =0 and $\Phi$ =2.5˚ for the sliding experiments. Solid lines represent the average behaviour during the experiments**
**at the same slope, whereas the grey areas show the standard deviation of the variable considered.**

### 4.2.2 Pendulum oscillations

To further investigate the sensor ability to capture rotation, the cobble was used as a weight of a pendulum hanging from a

supporting frame (Figure 9b). An eye-hook screw was mounted on the top of the sensor enclosure and poked out of the

cobble top once sealed with the blue tack. A nylon thread was attached to the weight to allow vertical oscillations. The

length of the pendulum L was equal to 2.05 m. A grid panel was placed in the background to keep track of the pendulum

oscillations in the vertical plane. The weight position was tracked by a frontal GoPro camera using the detection algorithm

described in subsection 2.3. The sensor embedded in the cobble had the same settings as the previous experiments except for

the movement threshold. Since the oscillations were slow and collision-free, the accelerometer threshold was not sensitive

enough to activate the recording of the 9-axis sensor. For this reason, the oscillation experiments were carried out with no

movement threshold with the sensor recording as soon as the battery was connected. The cobble was placed at an initial

angle of 19˚ to the vertical and then allowed to oscillate for 4 periods. In this case, the run was repeated 5 times. The small

oscillation of a pendulum hanging from a massless string of length *L* and fixed at a frictionless pivot point is defined as (e.g.,

Kamberaj, 2021)





$$\theta = A_\theta \cos(\Omega t + \varphi) \tag{5}$$

where $\theta$ is the angle the weight of the pendulum forms with the vertical ($\sin\theta \approx \theta$), $\Omega$ is angular frequency equal to $\sqrt{\frac{g}{L}}$.

The coefficients $\varphi$ and $A_\theta$ for a standing start are equal to


$$\varphi = 0°; \ A_\theta = \theta_0 \tag{6}$$

where $\theta_0$ is the initial orientation. Position, velocity, and acceleration in the cartesian plane (y, z) are computed using the following formulas:

$$y = -L \sin\theta \qquad\qquad z = L(1 - \cos\theta) \tag{7,8}$$
$$v_y = v_t \cos\theta \qquad\qquad v_z = v_t \sin\theta \tag{9,10}$$
$$a_y = a_t \cos\theta + a_c \sin\theta \qquad\qquad a_z = v_t \sin\theta + a_c \cos\theta \tag{11,12}$$

where $v_t$ is the tangential velocity ($L\frac{d\theta}{dt}$), $a_t$ is the tangential acceleration ($L\frac{d^2\theta}{dt^2}$) and $a_c$ is the centripetal acceleration ($L\left(\frac{d\theta}{dt}\right)^2$). The experimental data are compared to the modelling results in Figures 12 and 13.

The pendulum trajectory is represented in Figure 12a. The average trajectories lie around 1.5 cm above the modelling prediction at the origin of the horizontal axis, whereas the experimental results match the theoretical values more closely at the edges of weight trajectories. The pendulum weight reaches a slightly higher elevation on the right-hand side making the experimental trajectory look somewhat asymmetric and off-centred. This notwithstanding, overall, the experimental trajectories match the modelling results reasonably well when the pendulum length and the initial angle are set at 2.05 m and 19˚, respectively. The experimental pitch angle is compared to the theoretical angular displacement in Figure 11b. Locally, the discrepancies are clear. The modelling prediction overestimates the average pitch angle. Moreover, experimental oscillations do not show a periodicity similar to the pendulum conceptual model. However, the experimental pitch angle ranges between -17.5˚ and 13.5˚ which is in good agreement the modelling angular displacement ranging between -19˚ and 19˚. These discrepancies propagate to the velocity and acceleration computed in the Cartesian reference system (Figure 13). The experimental values for the horizontal velocity vy are in good agreement with the modelling results (Figure 13a). Specifically, the average values match the theoretical ones and show a similar periodicity. Conversely, for the vertical velocity vz, the modelling framework well approximates the experimental periodicity but overestimates the velocity range (Figure 13b). The theoretical acceleration shows a periodicity similar to the experiments, roughly approximating the horizontal acceleration ay (Figure 13c) and underestimating the acceleration in the vertical direction az (Figure 13d).



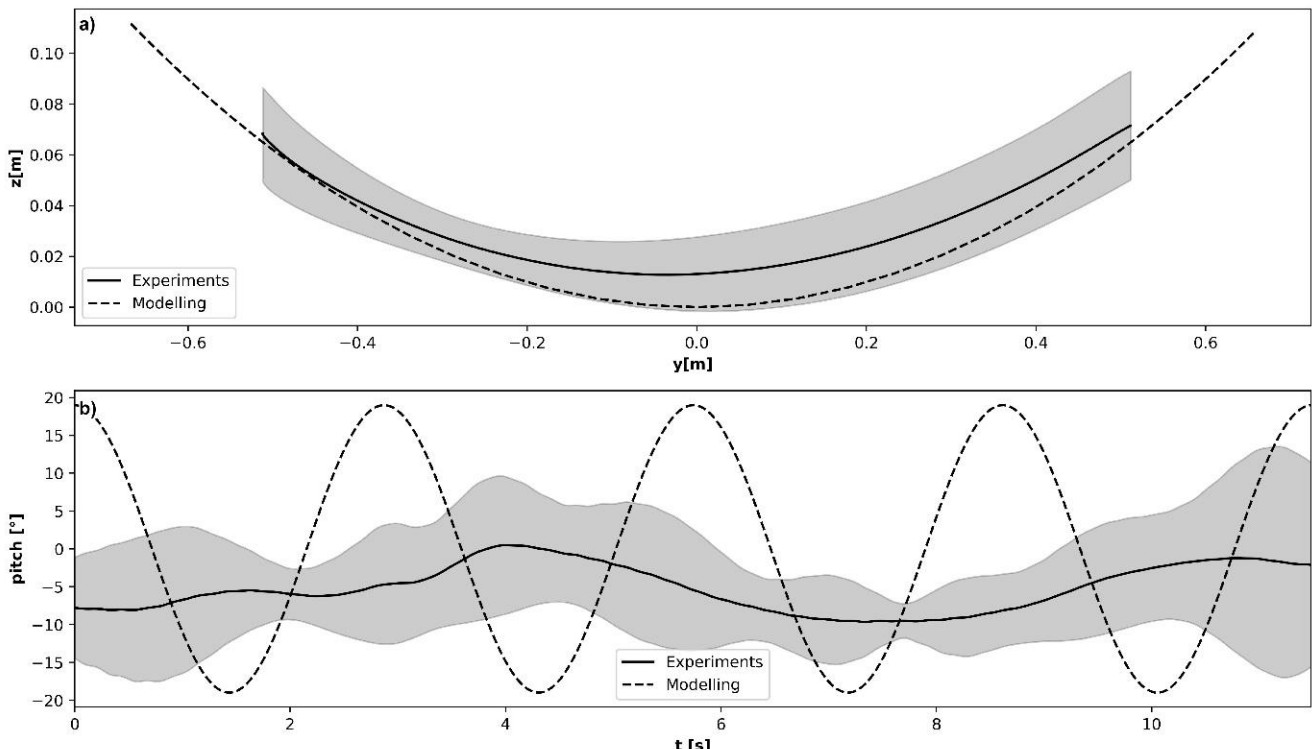

**Figure 12.** Comparison between the experiments and the analytical model. (a) Trajectories in the vertical plane z-y. (b) The angle the weight of the pendulum forms to the vertical. The response of the conceptual model is obtained for L = 2.05 m and $\vartheta_0$= 19°.






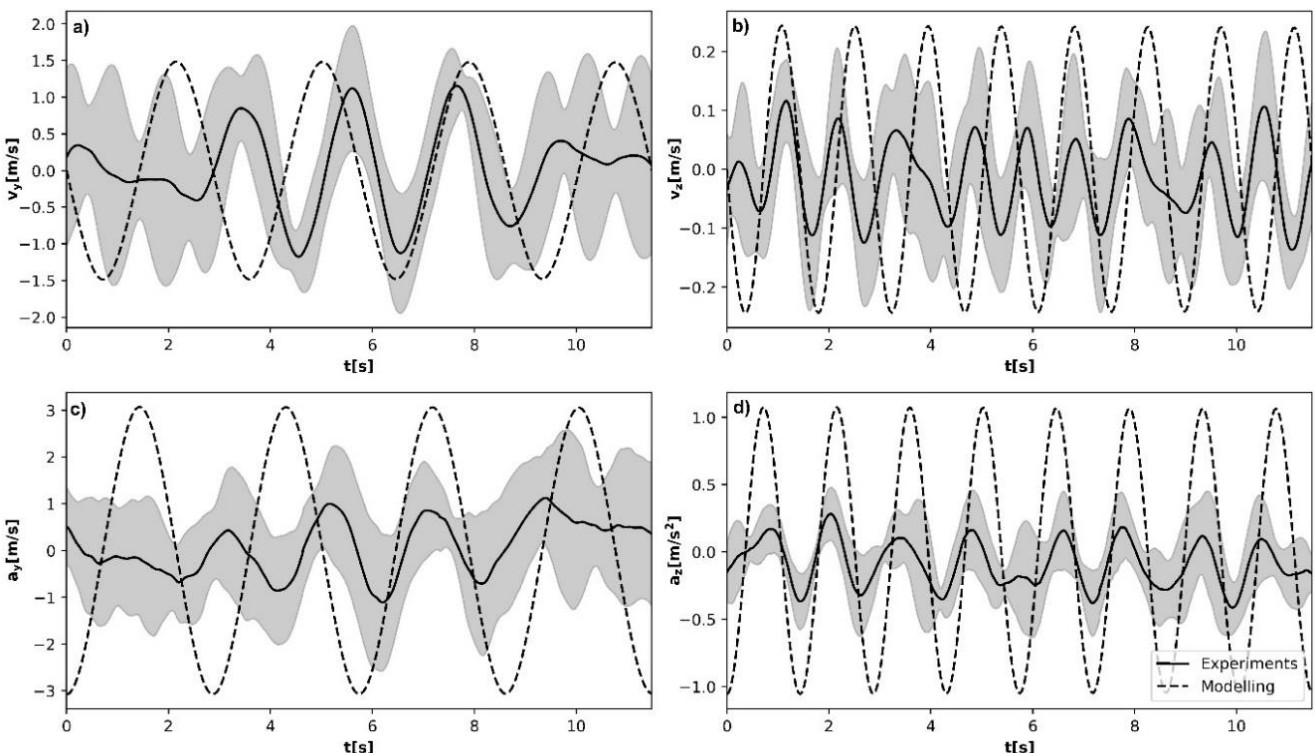

**Figure 13. Comparison between the experimental results and the analytical model for pendulum oscillations when the pendulum weight is released at 19°. (a) Horizontal velocity of the pendulum $v_y$. (b) Vertical velocity $v_z$. (c) Horizontal acceleration of the pendulum $a_y$. (d) Vertical acceleration $a_z$. Dashed lines show the response of the conceptual model for L = 2.05 m and $\vartheta_0$= 19°. Solid lines represent the average behaviour during the experiments, whereas the grey area shows the standard deviation of the variable considered.**


Some discrepancies are present in the comparison with the modelling framework, preventing the use of the sensors to infer the full kinematics in the proposed application. However, the sensor can discern between sliding and rolling. Furthermore, the range of velocity and acceleration in the experiments appear to be in good agreement with modelling results (with an uncertainty related to non-perfectly fixed position of the sensor within the borehole).


### 4.3 Limitations of the study

The results and the limitations evident from the experimental study give some suggestions on the use of the present smart 9-axis sensor for more advanced lab or field applications. The proposed data-fusion approach approximately describes movements for a cobble travelling down a tilting slope. Indeed, the range of estimated movements well represent the range predicted by conceptual models. However, the cobble movements as reconstructed in the sensor fusion approach are not completely matched by the conceptual models. Moreover, the variation shown by the kinetic energy as the slope changes are not fully explicable.   The limitations of the present study are briefly discussed below. First, the vertical elevation in camera-derived positions is computed assuming a point of contact with the board. When the table inclination is small (<45°), the



point of contact is ensured. However, for higher inclinations, the cobble starts losing contact with board. Thus, the bias in the vertical elevation may propagate into the variables calculated with respect to the vertical and horizontal axes. Second, despite capturing some essential features of the movements, the modelling frameworks show some deviations from the average experimental behaviour derived for the position, velocity, acceleration, and angle orientation. These deviations are possibly caused by the uneven surface and irregular mass distribution of the cobble (due to the borehole) that create a larger
variability of its path down the slope.

Regarding lab applications, using the present approach to study granular flows (i.e., of multiple grains) in the lab poses some problems. A sensor recording the movement can be installed in each cobble in the granular flow. However, cameras are not able to track a cobble when covered by the other cobbles forming the granular flow, and thus camera-based positions are not
available to constrain cobble motion. Without position measurements, the data-fusion approach used here is not applicable. Moreover, in collisional experiments with two cobbles, the present accelerometer showed some values exceeding the acceleration range. Hence, the range for the 9-axis sensor needs to be large enough in collisional experiments to collect meaningful data of the impacts (e.g., Maniatis et al., 2023).

Regarding field applications, the landslide topography is complex and variable and thus, even using all the data recorded by
the sensor in the field (namely acceleration, angular velocity, magnetic field, and GPS position), the linear Kalman filter used here is unlikely to appropriately reconstruct the boulder movements. Further investigations have to be dedicated to exploring the combined use of the IMU, magnetometers and embedded GPS to characterise landslide movements (Roskilly et al., 2022, 2023).

Another point to consider is that LoRaWAN signal propagation is sensitive not only to sand layers, as in the present study,
but also to weather conditions (temperature, humidity; Goldoni et al., 2022) and the distance between the gateway and network end-nodes (e.g., Goldoni et al., 2019; Savazzi et al., 2019). Some experiments were carried out to assess the LoRaWAN technology limitations by studying the RSSI distance decay in different environments (e.g., Goldoni et al., 2019; Ferreira et al., 2020). However, further investigations are needed to thoroughly characterise the LoRaWAN performance in landslide sites. Furthermore, to expand the use of smart sensors for early-warning system applications, LoRaWAN
technology has to provide low-latency data transmission with respect to the timescale of the events being monitored. Thus, the time lag between sensor recording and data retrieval should be carefully studied in field sites. To allow timely data transmission for early-warning system applications, it would be better to characterise the motion of the boulder by computing a high-level metric on-board the sensor (such as the total kinetic energy proposed in this study) and then send this via LoRaWAN with highest priority. Thus, by providing information on the activity of boulders distributed on a landslide, it
would be possible to analyse landslide movements in the time domain. Boulder activity would possibly help to map the boundaries of zones where displacements occur (e.g., Dini et al., 2021). Field activities have been carried out to further explore the potential of smart boulders in landslide hazard assessment (Roskilly et al., 2022, 2023).



Smart boulders then could provide localised ground-based measurements on the activity occurring on the landslide surface. However, understanding landslide behaviour requires monitoring data (at different spatio-temporal scales) to track local and global movements, correlate movements occurring on the landslide surface and the deeper layers, and get insight on the subsurface network sensitive to rainfalls. In the past, an integrated multi-sensor approach has been used to study landslide behaviour (e.g., Castagnetti et al., 2013, Casagli et al., 2017; Wang Z. et al., 2022). Hence, smart boulders could be integrated into a multi-sensor monitoring approach combining remote-sensing and ground-based measurements to capture fully the landslide behaviour.

## 5 Conclusions

Experiments here aimed to test smart sensors by studying separately their data transmission potential and sensor potential. The study investigated how LoRaWAN data transmission is affected when the sensor is covered by sand layers of different thickness. The study also investigated how to use the 9-axis sensor to detect the activity and mode of movements of a cobble travelling down a slope. The main experimental findings can be summarised in the following points.

- First, the Received Signal Strength Indicator (RSSI) is sensitive to the thickness of the sand layer covering the sensor. Moreover, RSSI shows more sensitivity to sand coverage in the magnetometer packets than in the gyroscope and accelerometer packets. Conversely, the Signal-to-Noise Ratio (SNR) stays approximately constant regardless of the sensor. The sensitivity of the LoRaWAN system to possible sand coverage adds a degree of complexity to wireless data transmission in the field and deserves further investigation to better frame the technology potential in early warning systems.

- Second, raw data allow detecting movement and separating two modes of movement, namely rolling, and sliding. However, raw data do not give reliable values for the acceleration, angular velocity, and magnetic field but they can evaluate approximately the magnitude of the movement.

- Third, by combining sensor data and camera-based data, it was possible to derive a full characterization of the movements of the cobble. A data fusion approach makes it possible to derive values for position, orientation, velocity, and acceleration whose overall range was approximately validated by simple theoretical frameworks.

Overall, smart sensors showed their potential and limitations to give new insights on the dynamics of complex hazardous flow. In field applications, it is challenging to apply the data fusion approach and so smart boulders can for now give us an indication of the presence of movement and a quantitative approximation of its intensity. At the same time, they could provide useful information when part of a multi-sensor network.

**Data Availability Statement.** The data used in this study are accessible upon request by contacting Alessandro Sgarabotto (alessandro.sgarabotto@plymouth.ac.uk). The DOI link with access to data still needs to be created.



**Author contributions.** AS and IM designed the lab experiments. AS carried out the experiments and led the data analysis and modelling with input from KR, CL, and IM. MC designed and molded the concrete hollow cobble for the experiments

with input from GLB. KR, GLB and AMAF contributed to the sensor development. AS and IM wrote up the manuscript, which was edited and reviewed by KR, CL, GLB, MC, and AMAF. IM, GLB and AMAF conceived the project and got the project funded.

**Competing interests.** The authors declare that they have no conflict of interest.


**Acknowledgments.** This research was supported by Natural Environment Research Council (grant no. NE/V003402/1, SENSUM: smart SENSing of landscapes Undergoing hazardous hydrogeologic Movement).

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
