# Peer review of "Supporting information of "Evaluating the use of smart sensors in ground-based monitoring of landslide movement with laboratory experiments""

_EGUsphere, 2023_

## Author Comment (AC1)

**RESPONSE TO THE COMMENTS ON Ms. Preprint egusphere-2023-2596**

Please note that in this rebuttal, reviewers' comments are denoted in *grey*, our detailed response is in black, and the new text of the revised version is in *italics*. Two different fonts are used for the reviewer's comments and the detailed response.

**Response to REVIEWER #1**

*Thus, the general recommendation is **reconsidered after major revisions.***

*Major Comments The table-top experiment in your manuscript presents some interesting ideas, notably the investigation of LoRaWAN signal strength propagation through sand. However, the approach to comparing results with classical equations of motion (EOMs) comes across as unnecessarily grandiose. Additionally, the pendulum analysis comes utterly short, as a single harmonic oscillator is deemed as model comparison, while the data clearly shows a beating effect. A significant oversight is the lack of discussion on scalability, especially pertinent given the context of smart sensors in landslide applications. In conclusion, substantial revisions are needed, particularly in data analysis for the pendulum section and a more grounded portrayal of the comparison to classical EOMs.*

**Major points**

1) *However, the approach to comparing results with classical equations of motion (EOMs) comes across as unnecessarily grandiose.*

   Response

   Thank you to the reviewer for this comment on the comparison with EOMs. We gave particular importance to this aspect because of the lack of experimental work on testing a similar sensor to track boulders embedded in slow-moving landslides. Besides MEMS sensor application on rockfalls (e.g. Niklaus et al., 2017; Caviezel et al., 2018, 2019; Noël et al., 2023), this sensor application on boulders embedded in landslides was discussed only in Dini et al. (2021) where no experiments were carried out. Dini et al., 2021 managed to determine the timing of movement but not the style or magnitude of movement as they were limited to just an accelerometer (i.e. not full IMU). In recent laboratory investigations published on NHESS (Dost et al., 2020), a similar sensor was installed in five pebbles travelling within a granular flow and no validation was proposed.

   The present study aims to test a sensor to track cobble motion down an inclined plane as a preparatory investigation to monitor boulders embedded in slow-moving landslides. Sensors are thus used to find an overall estimate of the magnitude of the movement. The comparison with EOMs was necessary to have an exploratory validation of the overall movement reconstructed by combining sensor data and camera-based position. By developing the camera-sensor fusion approach, the study aims to fill the gaps in previous experimental work on this topic and make the results and application of the sensors used more reliable for use in the field.

Although the sensor-based motion reconstruction does not match well motion equation predictions at each time step, the range of displacements, orientation angles, velocity, and acceleration in the experiments is similar to that predicted by the standard motion equation (Table 2).

**Table 2. Uncertainty estimation metrics for motion variables as derived from experiments on a 30⁰ inclined. Uncertainty estimation metrics. μ, σ the sample mean, and standard deviation of experiments as described by camera and sensors considering all repeats, respectively. Maximum and minimum values computed for motion equation predictions.**

| | | | y (m) | z (m) | $v_y$ (m/s) | $v_z$ (m/s) | $a_y$ (m/s$^2$) | $a_z$ (m/s$^2$) |
|---|---|---|---|---|---|---|---|---|
| **Rolling** | Experiments | $\mu$ | 1.933 | 0.096 | 1.040 | -0.378 | 0.025 | -2.977 |
| | | $\sigma$ | 0.181 | 0.019 | 0.410 | 0.220 | 2.285 | 2.586 |
| | Standard motion equation | Max | 3.7 | 0.547 | 2.714 | 0.0 | 3.287 | 0.0 |
| | | Min | 0.088 | 0.060 | 0.0 | -1.359 | -1.281 | -1.898 |
| **Sliding** | Experiments | $\mu$ | 0.786 | 0.229 | 0.974 | -0.150 | 3.161 | 5.246 |
| | | $\sigma$ | 0.080 | 0.025 | 0.379 | 0.158 | 5.701 | 1.567 |
| | Standard motion equation | Max | 0.547 | 0.547 | 2.965 | 0.0 | 3.927 | 0.0 |
| | | Min | 0.062 | 6.077 | 0.0 | -1.428 | -0.428 | -2.267 |

The elements detrimental to this comparison are the irregular object shape that does not hold in the conceptual model assumptions, the non-fixed position of the sensor within the sensor enclosure, and the borehole and the cotton-pad buffer. These limitations increase the degree of approximation in the sensor-based motion reconstruction and make the assumptions under which rigid motion equations are based not hold. This notwithstanding, the present sensor-based motion reconstruction represents an improvement in the understanding of how to use smart sensors for granular flow experiments.

The approach is different from previous MEMS applications in rockfalls tracking (e.g. Niklaus et al., 2017; Caviezel et al., 2018, 2019; Noël et al., 2023) and granular flow experiments (Dost et al., 2020). In rockfall MEMS application, videogrammetric trajectory is aided by an accelerometer and gyroscope. Position and velocity are inferred by videogrammetric trajectory, acceleration is used to characterise the impacts with the ground and the number of block rotations over a given number of video frames. Thus, acceleration and position are not combined in the fusion algorithm. Conversely, in Dost et al. (2020), 5 tagged pebbles embedded in granular flow experiments were tracked only by using the sensor installed in them without any position measurements. However, the reconstructed pebble trajectories did not all lie within the flume walls posing some questions on the reliability of sensor data in their own right. In granular flow experiments, the camera cannot track particles not visible, and thus a different framework must be developed to use smart sensors to study granular flows. Moreover, in Dost et al 2020, no orientation angles were computed, and no validation was carried out.

We added Table 2 to the discussion to have an estimate of the uncertainty in the experiments and better framed the novel use of the data-fusion approach in the context of boulders embedded in the body of a landslide.

2) *Additionally, the pendulum analysis comes utterly short, as a single harmonic oscillator is deemed as model comparison, while the data clearly shows a beating effect.*

Response

Thank you to the reviewer for highlighting the beating effect in the pendulum experiments. The effect is likely to be due to two different factors: nonuniform mass distribution of the cobble, and the lack of a rigid rod as a pendulum arm (a nylon thread was used instead). For this reason, the pendulum weight tends to slightly rotate out of the vertical plane introducing spurious rotations around different axes. Spurious rotations introduce additional mode frequencies in the pitch orientation angles causing the beating effect. The orientation angles were computed by feeding the accelerometers, gyroscopes, and magnetometer data in a filter developed by Mahoney et al. (2012). The filter described by Mahoney et al. (2012) was conceived to describe the orientation angles of a drone during flights. Besides the numerical error, the filter bias is related to a low-cost IMU sensor that is sensitive to vibrations and small movements. In the cobble, the sensor does not have a perfectly fixed position. The small movements are likely to increase the filter bias.

Thus, the data-fusion approach based on camera videos and sensor outputs fails to represent motion at each instant but manages to grasp the range of motion. The approximation in the motion reconstruction is ascribed to the conceptual model assumptions and some experimental conditions (cobble irregular shape, cobble nonuniform density distribution, nonfixed position of the sensor). This notwithstanding, given the potential to grasp the magnitude/range of motion, smart boulders embedded in a landslide could be used to get insight into the overall range of motion experienced in an event (Table 1). Thanks to the comment of the reviewer, we are adding more clarity to this aspect by adding Table 1 shown below to the updated version of the manuscript. The data shown in the table explain in more detail the evaluation of the uncertainty of the data-fusion approach compared to the standard motion equation.

**Table 1. Uncertainty estimation metrics for motion variables as derived from pendulum experiments. Uncertainty estimation metrics. μ, σ the sample mean, and standard deviation of experiments as described by camera and sensors considering all repeats, respectively. Maximum and minimum values are computed for motion equation predictions.**

| | | Pitch (°) | y (m) | z (m) | $v_y$ (m/s) | $v_z$ (m/s) | $a_y$ (m/s²) | $a_z$ (m/s²) |
|---|---|---|---|---|---|---|---|---|
| **Experiments** | $\mu$ | -0.090 | -0.064 | 0.040 | 0.028 | -0.016 | 0.083 | -0.087 |
| | $\sigma$ | 0.053 | 0.352 | 0.020 | 0.564 | 0.063 | 0.569 | 0.167 |
| **Standard motion equation** | Max | 0.331 | 0.658 | 0.112 | 1.487 | 1.487 | 3.076 | 1.078 |
| | Min | -0.332 | -0.667 | 0.0 | 0.244 | -1.487 | -3.076 | -1.059 |

We are planning to add pendulum experiment limitations to the discussion.

3) *A significant oversight is the lack of discussion on scalability, especially pertinent given the context of smart sensors in landslide applications.*

Response

Laboratory experiments were carried out using a single cobble and thus the effect of cobble size was beyond the scope of the experimental study. Data scalability surely can play an important role in using this sensing technology when cobble size increases or when multiple tagged cobbles/boulders stream data via LoRaWAN to the same gateway.

The effect of boulder shape on boulders in rockfall motion has been studied in laboratory experiments (e.g., Torsello et al., 2021) and field investigations (e.g., Caviezel et al., 2021). For kinematic and dynamic variables, scaling considerations were thoroughly discussed in Iverson (2015). However, no study has analysed the data scalability effect of shape on boulders embedded in the body of a landslide. A source of uncertainty in data scalability may arise from whether the boulder remains embedded in the body of a landslide for the entire duration of this motion changing the process type and hence its scaling. No previous study has shown how magnetometer data can change when the size of the object in which the sensor is installed increases. Thus, it is yet to be found how to proceed with scaling on magnetometer data. Moreover, the sensing network scalability depends on the number of devices streaming data to the gateway (Hart and Martinez, 2020). In the landslide sites under investigation with this technology, around 25-30 sensors are installed in boulders and all of them are connected to a single gateway (Roskilly et al., 2022, 2023). By increasing the number of gateways or the gateway bandwidth it is possible to cope with a network connecting more sensors and larger data size.

Thanks to the comments of the reviewer, we plan to add a subsection on data calability in the updated version of the manuscript.

4) *In conclusion, substantial revisions are needed, particularly in data analysis for the pendulum section and a more grounded portrayal of the comparison to classical EOMs.*

Response

We addressed this specific comment in points 1) and 2).

Caviezel, A., Ringenbach, A., Demmel, S.E., Dinneen, C. E., Krebs, N., Bühler, Y., Christen, M., Meyrat, G., Stoffel, A., Hafner, E., Eberhard, L. A., von Rickenbach, D., Simmler, K., Mayer, P., Niklaus, P. S., Birchler, T., Aebi, T., Cavigelli, L., Schaffner, M., Rickli, S.,

Schnetzler, C., Magno, M., Benini, L., Bartelt, P.: The relevance of rock shape over mass—implications for rockfall hazard assessments. *Nature Communication*, 12, 5546, 1-9, https://doi.org/10.1038/s41467-021-25794-y, 2021.

Dini, B., Bennett, G. L., Franco, A. M. A., Whitworth, M., Cook, K., Senn, A., and Reynolds, J.: Development of smart boulders to monitor mass movements via the Internet of Things: A pilot study in Nepal, *Earth Surface Dynamics*, 9(2), 295-315, https://doi.org/10.5194/esurf-9-295-2021, 2021.

Dost, B. J., Gronz, O., Casper, M. C., and Krein, A.: The potential of Smartstone probes in landslide experiments: How to read motion data, *Natural Hazards and Earth System Sciences*, 20(12), 3501-3519, https://doi.org/10.5194/nhess-20-3501-2020, 2020.

Hart, J. K., Martinez, K.: Sensor Networks and Geohazards. In Shroder, J. J. F. (Ed), *Treatise on Geomorphology* (Vol 1, pp. 100-120), Second Edition, Academic Press, https://doi.org/10.1016/B978-0-12-818234-5.00037-7, 2022.

Iverson, R. M.: Scaling and design of landslide and debris-flow experiments. *Geomorphology*, *244*, 9–20. https://doi.org/10.1016/j.geomorph.2015.02.033, 2015.

Mahony, R., Hamel, T., Morin, P., & Malis, E.: Nonlinear complementary filters on the special linear group, *IEEE Transactions on Automatic Control, Institute of Electrical and Electronics Engineers*, *53*(5), 1203–1217, https://doi.org/10.1080/00207179.2012.693951, 2008.

Noël, F., Nordang, S. F., Jaboyedoff, M., Travelletti, J., Matasci, B., Digout, M., Derron, M.-H., Caviezel, A., Hibert, C., Toe, D., Talib, M., Wyser, E., Bourrier, F., Toussaint, R., Malet, J.-P., and Locat, J.: Highly energetic rockfalls: back analysis of the 2015 event from the Mel de la Niva, Switzerland, *Landslides*, 20:1561-1582, https://doi.org/10.1007/s10346-023-02054-2, 2023.

Roskilly, K., Bennett, G., Curtis, R., Egedusevic, M., Jones, J., Whitworth, M., Dini, B., Luo, C., Manzella, I., and Franco, A.: SENSUM project, Smart SENSing of landscapes Undergoing hazardous hydrogeomorphic Movement, EGU General Assembly 2022, Vienna, Austria, 23-27 May 2022, EGU22-10289, https://doi.org/10.5194/egusphere-egu22-10289, 2022.

Roskilly, K., Bennett, G., Clark, M., Franco, A., Egedusevic, M., Curtis, R., Jones, J., Whitworth, M., Luo, C., and Manzella, I.: Smart cobbles and boulders for monitoring movement in rivers and on hillslopes, EGU General Assembly 2023, Vienna, Austria, 24-28 Apr 2023, EGU23-14870, https://doi.org/10.5194/egusphere-egu23-14870, 2023.

Torsello, G., Vallero, G., and Castelli, M.: The role of block shape and slenderness in the preliminary estimation of rockfall propagation, IOP Conference Series Earth Environment Science, 833, 012177, 1-9, https://doi.org/10.1088/1755-1315/833/1/012177, 2021

**Specific comments**

*Figure annotations: While there are strong opinions around for this one, feel free to stick with yours: mine would be "x (m) " instead of "x[m]". The [] brackets are in physics the unit operator used in dimensional analysis. Hence [x] = m, [mag] = G, and so on.*

Response

We now will use round brackets instead of square brackets in all figure annotations following the suggestion of the reviewer.

*L 94-97: Leave out the obvious. The committed reader will find out about the structure of the paper.*

Response

We are planning to rephrase that part at the end of the introduction following the reviewer's suggestion. The amended lines read as follows:

*"This study shows the results from LoRaWAN data transmission tests and the findings on raw and processed data for the cobble motion (Section 3). After comparing experimental results to standard motion equations (Section 4), the study discusses the strengths and weaknesses of the smart sensor technology in monitoring boulders and the challenges awaiting to be addressed to improve the technology (Section 5)."*

*L115-116: distinguish between a 9DOF IMU and a 3DOF accelerometer. Yes, the web page states the ST LIS2DH tracks motion, but it actually only tracks accelerations. Be precise.*

Response

In lines 115-116, the word "motion" was replaced with "acceleration" to make the sentence precise.

*"The device has an additional low-power 3-axis accelerometer sensor (ST LIS2DH) that monitors acceleration continuously."*

*L120 vs L 115: The introduction of the smart sensor as mini-GPS Tracker, and only stating later, that GPS is switched off, is confusing. Sensor experts wonder immediately, what use a GPS tracker is for an indoor experiment.*

Response

The specific name of the Miromico-manufactured sensor is "*mini-GPS Tracker*". However, we agree with the reviewer that in this context the sensor name is misleading since the GPS was deactivated in the indoor experiments. Thus, the sensor name was deleted to avoid confusion. The lines were rephrased as:

*"Specifically, the smart sensor used was a Miromico-manufactured device (Miromico manual, 2020a, b) equipped with a 9-axis sensor comprising accelerometers, gyroscopes, and magnetometers (ST LSM9DS1)."*

Moreover, in lines 119-120, "*and a GPS receiver (this was deactivated for indoor experiments)*" was deleted to avoid confusion.

*L115: Please add an image of the stripped down, commercially available Miromico to Figure 2.*

Response

We added an image of the sensor in Figure 2c and changed accordingly the figure caption. Please find below Figure 2 after the corrections.

[Figure]

**Figure 2. Laboratory experiments pictures. (a, b) Cobble rendering. (c) Sensor tag. (d, e, and f) Sensor installation in the cobble. The starting position of the cobble within the release box for (g) rolling experiments and (h) sliding experiments while embedded in a thin layer of sand.**

*Figure 4: Please add RSSI and SNR explanations for only image-readers. Increase resolution of images, they look pixelated.*

Response

We increased the figure resolution and added a brief explanation of RSSI and SNR in the caption. Please find below Figure 4 after the corrections.

[Figure]

**Figure 4. Cumulative probability distribution of (a and b) Received Signal Strength Indicator (RSSI) and (c, d) Signal Noise Ratio (SNR) when the signal is transmitted through different sand layer depths, namely 0 cm, 5 cm, 8 cm, and 10 cm. RSSI and SNR are measurements of the signal power received in the gateway and the ratio of the signal power and the noise power, respectively. (a, c) Accelerometer and gyroscope packets. (b, d) Magnetometer packets.**

*Figure 5: Increase dpi.*

Response

Thank you for this comment. The figure resolution was increased to 580 dpi. Moreover, I made the changes suggested by reviewer 2: the range of the x-axis is set equal in all subplots (a-f) (namely, 0.0 s - 2.8 s), the range of the y-axis is set equal in the corresponding subplots (a and d; b and e; c and f). Please find below Figure 5 with increased resolution.

[Figure]

**Figure 5. Raw recordings of the three sensor types on a 30° incline for (a, b and c) a rolling experiment and (d, e and f) a sliding experiment. (a, d) gyroscopes data, (b and e) accelerometers data. (c, f) Magnetometers data after upsampling. The solid line refers to the x axis, the dashed line to the y axis and the dotted line to the z axis. The solid red line shows the time when the cobble passes over the slope break.**

*Figure 6: Increase dpi. Legend with capital letters in Test 1, etc.*

Response

The resolution will be increased, and amendments will be made to the legend. Please find below Figure 6 after the corrections suggested.

[Figure]

**Figure 6. Trajectories extracted from camera videos for (a) rolling tests and (b) sliding tests carried out on a 30° incline. The rolling tests were repeated three times, the sliding tests were repeated four times. Each run is denoted by the different line styles. The solid red line defines the position of the slope break.**

*Equation 1: I have seen this before, the equation and the ascribing to a specific publication dated from this century. With all due respect for your work, as a respectful note to a fellow scientist, it's crucial to maintain academic integrity and historical accuracy in scientific publications. Attributing well-established scientific principles, such as the equation representing total mechanical energy, to contemporary authors overlooks the foundational work of early physicists like Isaac Newton and Leonhard Euler. Yes, we are talking the big names here. This not only misrepresents the origins of these fundamental concepts but also undermines the rich history of scientific discovery. It's important for all scientific literature to accurately reflect the development of these theories over time and give credit where it's historically due.*

Response

I made some changes using only book references to introduce classical physics equations. Total Kinetic energy equation (eq. (1)) is introduced using the reference "(e.g., Díaz, 2019)", whereas standard motion equations (eqs. (2) and (5)) are introduced using the reference "(e.g. Kamberaj, 2021)". Classical mechanics books introduce fundamental concepts acknowledging the work of the fathers of mechanics and frame their foundational work highlighting their importance within modern physics.

Díaz, E. O.: *3D Motion of Rigid Bodies - A Foundation for Robot Dynamics Analysis*. Springer Cham, pp. 474, https://doi.org/10.1007/978-3-030-04275-2, 2019.

Kamberaj, H.: *Classical mechanics*, Berlin, Boston: De Gruyter, https://doi.org/10.1515/9783110755824, 2021.

*Equation 2, 3 and 4: A rigid body travelling down a tilted plane and its associated equations of motion are a common high-school problem. While it is perfectly valid to restate these equations here – again, not really invented or proposed the first time by any individual in the 21st century - as they are applied in a specific context, such as the kinematics of a rigid body traveling down a tilted plane with varying curvature, a lengthy derivation is unnecessary.*

Response

We deleted equations (2)-(3) and (7)-(12) as they can be easily derived from standard motion equation (eqs. (2) and (5)). Thank you.

*Pendulum Oscillation: The derivation of the pendulum oscillations equations starting from line 465 in the document is a detailed and lengthy process. It appears to be a standard approach found in textbooks on classical mechanics. While thorough, the extended focus on these foundational equations might be considered excessive for a journal publication, where readers typically expect more concise presentations of well-established theories.*

Response

Equations of classical mechanics are stated to define the nomenclature used in the validation section. Equations (2)-(3) and (7)-(12) were added in the manuscript only for the sake of completeness. To cut the description of well-known equations and not indulge in their presentation, equations (2)-(3) and (7)-(12) were deleted.

*Fig. 12 and 13: The significant discrepancies observed in the comparison with the classical equations of motion are a critical issue. The noted beating in the signal is particularly concerning and suggests a fundamental problem that needs to be addressed. It's imperative for the authors to rigorously investigate the origins of these discrepancies. This deeper analysis is crucial for assessing the validity of using the sensor technology in the context of these standard motion equations and for ensuring the accuracy and reliability of the study's findings.*

Response

We addressed this question in point 2) above.

*General remark on "modelling": To clarify, comparing experimental results with classical equations of motion should not be termed "modelling" in a strict sense. This comparison is more accurately described as a validation or verification step, where the experimental data is tested against established theoretical principles. "Modelling" typically refers to the development of new theoretical frameworks or the application of existing theories to simulate complex systems, which is distinct from merely comparing data with standard motion equations. This distinction is important for accurately conveying the nature of the scientific work being done.*

Response

We agree with the reviewer's comments. To highlight the distinction between new theoretical frameworks and standard motion equations, we replaced the label "Modelling" with "Motion equation" in all figures. Moreover, in the text, we replaced "theoretical predictions/results" with "motion equation predictions/results" and "modelling predictions" with "motion equation predictions".

The data-fusion approach presented in the study enables describing even complex cobble motion (for example when the cobble bounces on the table). The fused solution gives more information than camera tracking or IMU alone. On the one hand, by using only inertia sensor data, acceleration, velocity, and position are not estimated well and by using only camera-based positions, orientation angles and angular velocity are not estimated and separating between rolling and sliding becomes less intuitive.

The averaged values of motion variables computed by the data fusion approach are in the same range as the corresponding variables computed by the standard motion equation (Tables 1 and 2). Thus, despite the cobble irregular shape, non-uniform mass distribution, and nonfixed position of the sensor embedded, the fused solution captures an overall movement that is physically based. However, by looking at how each motion variable changes over time, there are discrepancies. These deviations cannot be fully explained by standard motion equations, or the data fusion approach developed in the study. On the one hand, the standard motion equations are not ideal for thorough validation. Cobble motion down the slope is more complex than what is simply described by the standard motion equations that do not consider nonuniform mass distribution, the irregular shape, and thus different areas of contact with the plane. The cobble pendulum is not built using a metal rod and thus, since the mass distribution and shape are not regular, there are spurious rotations interfering with those on the vertical plane. Thus, assumptions under standard motion equations do not perfectly hold in the cases considered.

---

## Author Comment (AC2)

**RESPONSE TO THE COMMENTS ON Ms. Preprint egusphere-2023-2596**

Please note that in this rebuttal, reviewers' comments are denoted in *grey*, our detailed response is in black, and the new text of the revised version is in *italics*. Two different fonts are used for the reviewer's comments and the detailed response.

**Response to REVIEWER #2**

Referee's comments on the manuscript "**Evaluating the use of smart sensors in ground-based monitoring of landslide movement with laboratory experiments**"

*The submitted manuscript aims to derive long-term rock movements and landslides through in-situ mounted sensors. Various laboratory experiments were carried out for this purpose.*

*The long-term goal of in situ-based landslide detection is socially relevant, and its achievement is scientifically desirable. However, how this manuscript is presented is not (yet) relevant for publication in its present form. The applied methods are per se to be embraced (smart sensors and the fusion of their gyroscope data with those of the magnetometer and accelerometer), but the validation presented (e.g. Fig. 11-13) are too far from the compared mental model, namely the center of mass of a rigid body (=single point). This raises both the questions of whether (a) the measured data and its fusion chain are not consistent or (b) whether the applied model is suited for the used complex cobble. Usually, favorable laboratory conditions are chosen. This has not been the case here, as the damping through the cotton pads around the sensors (which should protect them) creates an environment that is not comparable with the rigid body model.*

*So, in my opinion, the manuscript and its current results are not yet usable for the NHESS readership. On the one hand, the same results should be compared with a new comparison model. Instead of the cobble motions, the different movements of the sensors on its cotton pad are recorded. Therefore, the manuscript better fits in a signal processing journal, especially as the embedding in the natural hazards literature is not without some reservations (see specific comments). On the other hand, the experiments could be repeated with a rigid sensor attachment to the rock. Then, the simple single-point model can be applied, and the outcome will align with the NHESS scope.*

Response

We thank the reviewer for their comments on the validation of the sensors in the lab for application in landslide movement tracking. We understand their reservation on the methods used here and we will try to explain the approach and the reasoning behind the submission of this manuscript to NHESS.

The use of MEMS sensors in earth science application is still at an early stage. Understanding how to effectively use these sensors for geohazard evaluation is of great interest and requires a cross-disciplinary effort. The potential and limitations of MEMS applications are of interest to all geoscientists who have been trying to use sensing technology on more and more lines of research (e.g. Mao et al., 2019; Hart, and Martinez, 2020; Wang et al., 2022). The novelty of the study is in the type of sensor, the phenomenon for which the sensor is tested in laboratory experiments, and the approach to couple sensor recordings and camera-based positions.

Aim. The present study is part of a larger project named SENSUM (smart SENSing of landscapes Undergoing hazardous hydrogeologic Movement). The aim of the research on SENSUM and by extension in this paper was to test a sensing technology to estimate the timing of hazardous

movement, the magnitude of movement and the mode of movement of boulders embedded in slow-moving landslides. Moreover, the research on SENSUM wanted to test the transmission of movement-related data via LoRaWAN. The research project did not aim to measure any impact forces. The research goals on SENSUM are thus different from those in the existing body of work which performed more controlled experiments and wanted to measure physics and forces of processes such as rockfalls (Niklaus et al., 2017; Caviezel et al., 2018, 2019; Noël et al., 2023) or sediment transport (Maniatis et al., 2023). Regarding the motion of boulders embedded in a slow-moving landslide, Dini et al. (2021) managed to determine the timing of movement of using LoRaWAN but not the style or magnitude of movement as the sensor used was composed of just an accelerometer (i.e. not full IMU). The sensor used on SENSUM was a 9-axis device equipped with accelerometers (16-g range), gyroscopes (2000 °/s range), and magnetometers (16-Gauss range) that start recording when the acceleration exceeds a custom-defined threshold. This experimental study aimed to test this sensor to track cobble motion down an inclined plane as a preparatory investigation to monitor boulders embedded in slow-moving landslides. Sensors were thus used to find an overall estimate of the magnitude of the movement and understand the mode of movement. Moreover, the data transmission through LoRaWAN was tested under different thicknesses of sand layers to investigate how sand medium can affect data sending as it has not been tested before. Hence, beside trying to match with equation of motion, there are other ways the experimental data can be useful for the use of the sensor in field applications in slow-moving landslides.

Approach. In the present study, camera-based position and sensor data are fused into the Kalman filter. Hence, the approach is different from previous MEMS applications in rockfalls tracking (e.g. Niklaus et al., 2017; Caviezel et al., 2018, 2019; Noël et al., 2023) and granular flow experiments (Dost et al., 2020). In rockfall MEMS application, videogrammetric trajectory is aided by an accelerometer and gyroscope. Position and velocity are inferred by videogrammetric trajectory, acceleration is used to characterise the impacts with the ground and the number of block rotations over a given number of video frames. In recent granular flow experiments published on NHESS (Dost et al., 2020), the pebbles are not tracked by the camera since they are buried within the granular flow. Thus, in research papers mentioned, acceleration and position are not combined in the fusion algorithm.

Sensor. The sensor used in the present study is similar to that used in granular flow experiments (Dost et al., 2020), cobble tracking (Gronz et al., 2016), and debris tracking (Spreizter et al, 2019) in laboratory flume experiments. Differently from rockfall MEMS applications (Volkwein and Klette, 2014; Niklaus et al., 2017; Caviezel et al., 2018, 2019; Noël et al., 2023), the sensor used on SENSUM and by extension in the present work does not have a high-range IMU since the research does not aim to measure forces or study the impacts on the ground.

In industrial applications, there are different ways to clamp a sensor to a machine (e.g., stud, magnetic, wax, and adhesive mounting; Ewins 2000 - chp3). The sensor accuracy on high-frequency signals depends on the rigid attachment between the sensors and the object to monitor (e.g., Ewins 2000 - chp3). Indeed, the stiff mechanical connection between the inertial sensor and the object ensures that the motion of both bodies at all frequencies is the same. Therefore, any inertial force applied to the object is transmitted to the sensor due to the rigid attachment. High inertial force may require increasing the sensor range making the sensor sensitivity decrease (Dini et al., 2021). However, under high inertial (impact) forces, regardless of the range, MEMS sensors are likely not to be robust enough

to withstand the impact and thus they can break apart or malfunction (Feng et al., 2023). To prevent damage to the sensors, soft buffers can dampen the impact overload and preserve the integrity of the sensors affecting its accuracy (Feng et al., 2023). Hence, the rigid attachment would be ideal for a comparison with a reference (numerical model, theoretical model, standard motion equation), but it is not ideal for sensor integrity.

Despite the limitations related to the cotton pad buffer and the nonfixed position of the sensor embedded in the cobble, the averaged values of motion variables computed by the data-fusion approach are within the range predicted by the standard motion equation (Table 2).

**Table 2. Uncertainty estimation metrics for motion variables as derived from experiments on a 30⁰ inclined. Uncertainty estimation metrics. μ, σ the sample mean, and standard deviation of experiments as described by camera and sensors considering all repeats, respectively. Maximum and minimum values computed for motion equation predictions.**

| | | | y (m) | z (m) | $v_y$ (m/s) | $v_z$ (m/s) | $a_y$ (m/s$^2$) | $a_z$ (m/s$^2$) |
|---|---|---|---|---|---|---|---|---|
| **Rolling** | **Experiments** | $\mu$ | 1.933 | 0.096 | 1.040 | -0.378 | 0.025 | -2.977 |
| | | $\sigma$ | 0.181 | 0.019 | 0.410 | 0.220 | 2.285 | 2.586 |
| | **Standard motion equation** | Max | 3.7 | 0.547 | 2.714 | 0.0 | 3.287 | 0.0 |
| | | Min | 0.088 | 0.060 | 0.0 | -1.359 | -1.281 | -1.898 |
| **Sliding** | **Experiments** | $\mu$ | 0.786 | 0.229 | 0.974 | -0.150 | 3.161 | 5.246 |
| | | $\sigma$ | 0.080 | 0.025 | 0.379 | 0.158 | 5.701 | 1.567 |
| | **Standard motion equation** | Max | 0.547 | 0.547 | 2.965 | 0.0 | 3.927 | 0.0 |
| | | Min | 0.062 | 6.077 | 0.0 | -1.428 | -0.428 | -2.267 |

The sensor integration involved in the filtering procedure described in this study addresses this problem. Recently, the sensor filtering procedure for granular flow experiments was published on this journal (Dost et al., 2020) without measuring position and providing validation. The present study builds on the filtering technique shown in their work and validate the overall motion magnitude using standard motion equations. Moreover, this study investigates the sensitivity of LoRaWAN data transmission to sand layers of different thicknesses which has never been studied before. The data transmission sensitivity to sand has important implications for the development and use of this technology in the field as part of the research on SENSUM. For these reasons, the present manuscript was not submitted to a sensor-focused journal. The method has some limitations that were openly discussed in the manuscript to favour a possible technological advancement and a scientific discussion. Despite the limitations indicated in the discussion, the range of camera-sensor motion is similar to that of the conceptual model.

Table 2 shown above was added to the updated version of the manuscript to improve the evaluation of the uncertainty of the data-fusion approach compared to the standard motion equation. Thanks to the reviewer's comment, the limitations related to the cotton pad buffer and the nonfixed position of the sensor embedded in the cobble are planned to be added to the discussion.

Caviezel, A., Schaffner, M., Cavigelli, L., Niklaus, P., Bühler, Y., Bartelt, P., Magno, M., and Benini, L.: Design and evaluation of a low-power sensor device for induced rockfall experiments, IEEE

Transactions on Instrumentation and Measurement, 67(4), 767-779, https://doi.org/10.1109/TIM.2017.2770799, 2018.

Caviezel, A., Demmel, S. E., Ringenbach, A., Bühler, Y., Lu, G., Christen, M., Dinneen, C. E., Eberhard, L. A., von Rickenbach, D., and Bartelt, P.: Reconstruction of four-dimensional rockfall trajectories using remote sensing and rock-based accelerometers and gyroscopes, *Earth Surface Dynamics*, 7(1), 199-210, https://doi.org/10.5194/esurf-7-199-2019, 2019.

Dini, B., Bennett, G. L., Franco, A. M. A., Whitworth, M., Cook, K., Senn, A., and Reynolds, J.: Development of smart boulders to monitor mass movements via the Internet of Things: A pilot study in Nepal, *Earth Surface Dynamics*, 9(2), 295-315, https://doi.org/10.5194/esurf-9-295-2021, 2021.

Dost, B. J., Gronz, O., Casper, M. C., and Krein, A.: The potential of Smartstone probes in landslide experiments: How to read motion data, *Natural Hazards and Earth System Sciences*, 20(12), 3501-3519, https://doi.org/10.5194/nhess-20-3501-2020, 2020.

Ewins, D. J.: *Modal Testing: Theory, Practice and Application*, Second Edition, Research Studies Press, Ltd. Ltd., Exeter (UK), ISBN:978-0-863-80218-8, 2000.

Feng, D., Shi, Y., Zhao, R., Chen, Y., Zhang, P., Guo, H., and Guo, T.: Study on the mechanism of buffer absorbing energy of double-layer heterostructure based on viscoelastic materials for MEMS devices, *Sensors and Actuators: A. Physical*, 364, 114190, https://doi.org/10.1016/j.sna.2023.114790, 2023.

Gronz, O., Hiller, P. H., Wirtz, S., Becker, K., Iserloh, T., Seeger, M., Brings, C., Aberle, J., Casper, M. C., and Ries, J. B.: Smartstones: A small 9-axis sensor implanted in stones to track their movements, *Catena*, *142*, 245–251, https://doi.org/10.1016/j.catena.2016.03.030, 2016.

Hart, J. K., and Martinez, K.: Sensor Networks and Geohazards. In Shroder, J. J. F. (Ed), *Treatise on Geomorphology* (Vol 1, pp. 100-120), Second Edition, Academic Press, https://doi.org/10.1016/B978-0-12-818234-5.00037-7, 2022.

Maniatis, G., Toming G., Hoey, T. B., and Tuhtan, J.: KIVI the smartest of pebbles, *Proceedings of the 40th IAHR World Congress*, 21-25 August, Vienna, Austria, https://doi.org/10.3850/978-90-833476-1-5_iahr40wc-p0904-cd, 2023.

Mao, F., Khamis, K., Krause, S., Clark, J., and Hannah, D. M.: Low-Cost Environmental Sensor Networks: Recent Advances and Future Directions, *Frontiers in Earth Science*, 7, 221, 1-8, https://doi.org/10.3389/feart.2019.00221, 2019.

Niklaus, P., Birchler, T., Aebi, T., Schaffner, M., Cavigelli, L., Caviezel, A., Magno, M., and Benini, L.: Stonenode: A low-power sensor device for induced rockfall experiments, *2017 IEEE Sensors Applications Symposium (SAS)*, 1–6, https://doi.org/10.1109/SAS.2017.7894081, 2017.

Noël, F., Nordang, S. F., Jaboyedoff, M., Travelletti, J., Matasci, B., Digout, M., Derron, M.-H., Caviezel, A., Hibert, C., Toe, D., Talib, M., Wyser, E., Bourrier, F., Toussaint, R., Malet, J.-P., and Locat, J.: Highly energetic rockfalls: back analysis of the 2015 event from the Mel de la Niva, Switzerland, *Landslides*, 20:1561-1582, https://doi.org/10.1007/s10346-023-02054-2, 2023.

Spreitzer, G., Gibson, J., Tang, M., Tunnicliffe, J., and Friedrich, H.: SmartWood: Laboratory experiments for assessing the effectiveness of smart sensors for monitoring large wood movement behaviour, *Catena*, 182(July), https://doi.org/10.1016/j.catena.2019.104145, 2019.

Volkwein, A. and J. Klette, J.: Semi-automatic determination of rockfall trajectories, *Sensors*, 14, 10, 18187-18210, https://doi.org/10.3390/s141018187, 2014.

Wang, C., Guo, W., Yang, K., Wang, X., and Meng, Q.: Real-Time Monitoring System of Landslide

Based on LoRa Architecture, *Frontiers in Earth Science*, 10:899509, 1-11, https://doi.org/10.3389/feart.2022.899509, 2022.

**Specific comments**

*L13:*
*process type instead of hazard*

Response

We changed "*hazard*" with "*process type*" in the abstract. Thank you.

*L28:*
*Sim et al. 2022 are talking rather about risks, not the hazard.*

Response

The sentence was rephrased as (lines 27-29):

*"The higher frequency and intensity of extreme weather conditions under climate change have increased global landslide hazards (e.g., Gariano and Guzzetti, 2016) leading to the development of different approaches in landslide risk management (e.g., Sim et al., 2022)."*

*L34:*
*https://doi.org/10.1038/s43247-023-00909-z is also an important source here.*

Response

Thank you for bringing this very recent publications to us. I added it as you suggested in Line 34.

*L65/557*
*Citing EGU-General assembly abstracts is not the best practice.*

Response

Field investigations using MEMS sensors to monitor boulders embedded in a landslide and LoRAWAN to transmit data are rare because the technology has just started being deployed in geoscience applications. We mentioned the two most recent abstracts on this topic.

*L74-76:*
*The data of pressure sensors were barely used in these studies.*

Response

We wanted to highlight the sensors composing StoneNode (Niklaus et al., 2017; Caviezel et al., 2018, 2019) which are different from those composing the device used in our study. We rephrased the sentence as:

"*The sensor was equipped with accelerometers, gyroscopes, and a pressure sensor. The linear and angular motion were tracked by the accelerometers and the gyroscopes respectively. Both sensors helped to detect and characterise the collision with the ground. Conversely, the pressure sensor was used to measure altitude differences.*"

*L78:*
*The cited sources here make no sense: They were all published before the smart sensors publications (L74-76) and the mentioned 3D rockfall modeling approach was already validated with the publication by Dorren (2005).*

Response

True, the sentence is misleading and thus was rephrased as

*"This provides a new tool to collect a dataset to further improve and validate modelling frameworks on rock falls that had been previously calibrated only through case studies (Caviezel et al., 2018, Dorren et al., 2011; Dorren, 2016)."*

*L94-97:*
*This section is unnecessary. If you use meaningful sub-titles (as you do), the reader can easily navigate without this section. However, if you want to add this section, add at least the purpose and content of section 4.*

Response

Thank you for this comment. We rephrased that part at the end of the introduction. The amended lines read as follows (lines 97-100):

*"This study shows the results from LoRaWAN data transmission tests and the findings on raw and processed data for the cobble motion (Section 3). After comparing experimental results to simple conceptual models (Section 4), the study discusses the strengths and weaknesses of the smart sensor technology in monitoring boulders and the challenges awaiting to be addressed to improve the technology (Section 5). "*

*L107:*
*The squared, inclined board was hinged to the rectangular, horizontal board along its shorter side.*

Response

We changed the sentence following your suggestion. Thank you.

*L108:*
*Synchronized recordings to what?*

Response

True, the sentence is misleading. We rephrased the sentence as *"The GoPro camera was paired to a remote controller via Wi-Fi to start and stop the recording remotely"*.

*L155:*
*How many different training images?*
*Second, in each training image, ...*

Response

The training images were 390. I added this detail in the text. Then, I made the amendments in the sentence following your suggestion.

*"First, different training images of the object were collected (i.e. 390 images). The images were the ground truths to train the model. Second, in each training image, ..."*

*L162:*
*Which one is the suitable built-in function in OpenCV?*

Response

The rectification was carried out using the built-in Python functions *cv2.initUndistortRectifyMap* and *cv2.warpPerspective*. We plan to add this detail between brackets in the updated version of the manuscript.

*L165:*
*Missing "." after "occurred"*

Response

True. We corrected the oversight. Thank you.

*L228-234:*
*Belongs rather to the methods section.*

Response

We moved this part to the method section (lines 222-226) following your suggestion. Thank you.

*L248-250:*
*Belongs rather to the discussions section.*

Response

We moved this part to the method section (lines 575-578) following your suggestion. Thank you.

*L280, Fig. 5:*
*Better comparability if the x-axis comprises in all subplots (a-f) the same interval (e.g., 0.0 s – 2.8 s). The same would also be nice for the y-axis in the corresponding subplots (a and d; b and e; c and f).*

Response

We made changes to Figure 5 following your suggestion. Please find below the corrected figure.

[Figure]

**Figure 5. Raw recordings of the three sensor types on a 30° incline for (a, b and c) a rolling experiment and (d, e and f) a sliding experiment. (a, d) gyroscopes data, (b and e) accelerometers data. (c, f) Magnetometers data after upsampling. The solid line refers to the x axis, the dashed line to the y axis and the dotted line to the z axis. The solid red line shows the time when the cobble passes over the slope break.**

*L306-310:*
*A procedure description belongs to the methods section.*

Response

We moved this part to the method section (lines 236-240) following your suggestion. Thank you.

*L320:*
*"We speculate" belongs never in a result section.*

Response

We deleted the sentence from the result section. The concept is already described in the discussion section. Thank you.

*L324:*
*Wrong sub figures mentioned (7 c,d instead of 7b,c)*

Response

True. We corrected the oversight. Thank you.

*L365:*
*From where do you have the linear velocity? Pronounce it (again) that this is from the video footage analysis.*

Response

The linear velocity is not inferred from the video analysis, but it is computed by the Kalman filter fed by the camera-based position and the sensor-based linear acceleration. We rephrased the sentence to make this point clear. Thank you.

*"The linear velocity is computed through the Kalman filter fed by the camera-based positions and sensor-based linear accelerations, whereas the angular velocity is evaluated from the orientation angles (Section 2.3, Figure 3)."*